# Sample Complexity Bounds for Score-Matching: Causal Discovery and Generative Modeling

**Zhenyu Zhu**[†], **Francesco Locatello**[‡*], **Volkan Cevher**[†*]

† École Polytechnique Fédérale de Lausanne     ‡ Institute of Science and Technology Austria

†{zhenyu.zhu, volkan.cevher}@epfl.ch    ‡francesco.locatello@ista.ac.at

## Abstract

This paper provides statistical sample complexity bounds for score-matching and its applications in causal discovery. We demonstrate that accurate estimation of the score function is achievable by training a standard deep ReLU neural network using stochastic gradient descent. We establish bounds on the error rate of recovering causal relationships using the score-matching-based causal discovery method of Rolland et al. [2022], assuming a sufficiently good estimation of the score function. Finally, we analyze the upper bound of score-matching estimation within the score-based generative modeling, which has been applied for causal discovery but is also of independent interest within the domain of generative models.

## 1 Introduction

Score matching Hyvärinen [2005], an alternative to the maximum likelihood principle for unnormalized probability density models with intractable partition functions, has recently emerged as a new state-of-the-art approach that leverages machine learning for scalable and accurate causal discovery from observational data Rolland et al. [2022]. However, the theoretical analysis and guarantees in the finite sample regime are underexplored for causal discovery even beyond score-matching approaches.

**Contributions:** In this work, we give the first sample complexity error bounds for score-matching using deep ReLU neural networks. With this, we obtain the first upper bound on the error rate of the method proposed by Rolland et al. [2022] to learn the topological ordering of a causal model from observational data. Thanks to the wide applicability of score-matching in machine learning, we also discuss applications to the setting of score-based generative modeling. Our main contributions are:

1. We provide the analysis of sample complexity bound for the problem of score function estimation in causal discovery for non-linear additive Gaussian noise models which has a convergence rate of log n/n with respect to the number of data. Importantly, our results require only mild additional assumptions, namely that the non-linear relationships among the causal variables are bounded and that the score function is Lipschitz. To the best of our knowledge, this is the first work to provide sampling complexity bounds for this problem.

2. We provide the first analysis of the state-of-the-art topological ordering-based causal discovery method SCORE [Rolland et al., 2022] and provide a correctness guarantee for the obtained topological order. Our results demonstrate that the algorithm's error rate converges linearly with respect to the number of training data. Additionally, we establish a connection between the algorithm's error rate and the average second derivative (curvature) of the non-linear relationships among the causal variables, discussing the impact of the causal model's inherent characteristics on the algorithm's error rate in identification.

---

[*]Share the senior authorship

37th Conference on Neural Information Processing Systems (NeurIPS 2023).

3. We present sample complexity bounds for the score function estimation problem in the standard score-based generative modeling method, ScoreSDE [Song et al., 2021]. In contrast to previous results [Chen et al., 2023a], our bounds do not rely on the assumption of low-dimensional input data, and we extend the applicability of the model from a specific encoder-decoder network architecture to a general deep ReLU neural network.

**High-level motivation and background:** Causal discovery and causal inference refer to the process of inferring causation from data and reasoning about the effect of interventions. They are highly relevant in fields such as economics [Varian, 2016], biology [Sachs et al., 2005], and healthcare [Sanchez et al., 2022]. In particular, some causal discovery methods aim to recover the causal structure of a problem solely based on observational data.

The causal structure is typically represented as a directed acyclic graph (DAG), where each node is associated with a random variable, and each edge represents a causal mechanism between two variables. Learning such a model from data is known to be NP-hard [Chickering, 1996]. Traditional approaches involve testing for conditional independence between variables or optimizing goodness-of-fit measures to search the space of possible DAGs. However, these greedy combinatorial optimization methods are computationally expensive and difficult to extend to high-dimensional settings.

An alternative approach is to reframe the combinatorial search problem as a topological ordering task [Teyssier and Koller, 2012, Solus et al., 2021, Wang et al., 2021, Rolland et al., 2022, Montagna et al., 2023b,a, Sanchez et al., 2023], where nodes are ordered from leaf to root. This can significantly speed up the search process in the DAG space. Once a topological ordering is found, a feature selection algorithm can be used to prune potential causal relations between variables, resulting in a DAG.

Recently, Rolland et al. [2022] proposed the SCORE algorithm, which utilizes the Jacobian of the score function to perform topological ordering. By identifying which elements of the Jacobian matrix of the score function remain constant across all data points, leaf nodes can be iteratively identified and removed. This approach provides a systematic way to obtain the topological ordering and infer the causal relations within the entire model. This method has achieved state-of-the-art results on multiple tasks Rolland et al. [2022] and has been extended to improve scalability Montagna et al. [2023b] also using diffusion models Sanchez et al. [2023] and to non-Gaussian noise Montagna et al. [2023a]. Interestingly, these approaches separate the concerns of statistical estimation of the score function from the causal assumption used to infer the graph (e.g., non-linear mechanisms and additive Gaussian noise). This opens an opportunity to study the convergence properties of these algorithms in the finite data regime, which is generally under-explored in the causal discovery literature. In fact, if we had error bounds on the score estimate without additional complications from causal considerations, we could study their downstream effect when the score is used for causal discovery.

Unfortunately, this is far from trivial as the theoretical research on score matching lags behind its empirical success and progress would have far-reaching implications. Even beyond causal discovery, error bounds on the estimation of the score functions would be useful for score-based generative modeling (SGM) [Song and Ermon, 2019, Song et al., 2021]. These have achieved state-of-the-art performance in various tasks, including image generation [Dhariwal and Nichol, 2021] and audio synthesis [Kong et al., 2021]. There has been significant research investigating whether accurate score estimation implies that score-based generative modeling provably converges to the true data distribution in realistic settings [Chen et al., 2023b, Lee et al., 2022, 2023]. However, the error bound of score function estimation in the context of score-based generative modeling remains an unresolved issue due to the non-convex training dynamics of neural network optimization.

**Notations:** We use the shorthand $[n] := \{1, 2, \ldots, n\}$ for a positive integer $n$. We denote by $a(n) \lesssim b(n)$: there exists a positive constant $c$ independent of $n$ such that $a(n) \leqslant cb(n)$. The Gaussian distribution is $\mathcal{N}(\mu, \sigma^2)$ with the $\mu$ mean and the $\sigma^2$ variance. We follow the standard Bachmann–Landau notation in complexity theory e.g., $\mathcal{O}, o, \Omega$, and $\Theta$ for order notation. Due to space constraints, a detailed notation is deferred to Appendix A.

## 2 Preliminaries

As this paper concerns topics in score matching estimation, diffusion models, neural network theory, and causal discovery, we first introduce the background and problem setting of our work.

## 2.1 Score matching

For a probability density function $p(\boldsymbol{x})$, we call the score function the gradient of the log density with respect to the data $\boldsymbol{x}$. To estimate the score function $\nabla \log p(\boldsymbol{x})$, we can minimize the $\ell_2$ loss over the function space $\mathcal{S}$.

$$\min_{\boldsymbol{s}\in\mathcal{S}} \mathbb{E}_p[\|\boldsymbol{s}(\boldsymbol{x}) - \nabla \log p(\boldsymbol{x})\|^2], \quad \hat{\boldsymbol{s}} = \arg\min_{\boldsymbol{s}\in\mathcal{S}} \mathbb{E}_p[\|\boldsymbol{s}(\boldsymbol{x}) - \nabla \log p(\boldsymbol{x})\|^2].$$

The corresponding objective function to be minimized is the expected squared error between the true score function and the neural network:

$$J_{\text{ESM}}(\boldsymbol{s}, p(\boldsymbol{x})) = \mathbb{E}_{p(\boldsymbol{x})}\left[\frac{1}{2}\left\|\boldsymbol{s}(\boldsymbol{x}) - \frac{\partial \log p(\boldsymbol{x})}{\partial \boldsymbol{x}}\right\|^2\right], \tag{1}$$

We refer to this formulation as explicit score matching (ESM).

Denoising score matching (DSM) is proposed by Vincent [2011] to convert the inference of the score function in ESM into the inference of the random noise and avoid the computing of the second derivative. For the sampled data $\boldsymbol{x}$, $\hat{\boldsymbol{x}}$ is obtained by adding unit Gaussian noise to $\boldsymbol{x}$. i.e. $\hat{\boldsymbol{x}} = \boldsymbol{x} + \boldsymbol{\epsilon}$, $\boldsymbol{\epsilon} \sim \mathcal{N}(0, \sigma^2 \boldsymbol{I})$. We can derive the conditional probability distribution and its score function:

$$p(\hat{\boldsymbol{x}}|\boldsymbol{x}) = \frac{1}{(2\pi)^{d/2}\sigma^d} \exp(-\frac{\|\boldsymbol{x} - \hat{\boldsymbol{x}}\|^2}{2\sigma^2}), \qquad \frac{\partial \log p(\hat{\boldsymbol{x}}|\boldsymbol{x})}{\partial \hat{\boldsymbol{x}}} = \frac{\boldsymbol{x} - \hat{\boldsymbol{x}}}{\sigma^2}.$$

Then the DSM is defined by:

$$J_{\text{DSM}}(\boldsymbol{s}, p(\boldsymbol{x}, \hat{\boldsymbol{x}})) = \mathbb{E}_{p(\boldsymbol{x},\hat{\boldsymbol{x}})}\left[\frac{1}{2}\left\|\boldsymbol{s}\big(\hat{\boldsymbol{x}} - \frac{\partial \log p(\hat{\boldsymbol{x}}|\boldsymbol{x})}{\partial \hat{\boldsymbol{x}}}\big)\right\|^2\right] = \mathbb{E}_{p(\boldsymbol{x},\hat{\boldsymbol{x}})}\left[\frac{1}{2}\left\|\boldsymbol{s}(\hat{\boldsymbol{x}}) - \frac{\boldsymbol{x} - \hat{\boldsymbol{x}}}{\sigma^2}\right\|^2\right]. \tag{2}$$

Vincent [2011] have proven that minimizing DSM is equivalent to minimizing ESM and does not depend on the particular form of $p(\hat{\boldsymbol{x}}|\boldsymbol{x})$ or $p(\boldsymbol{x})$.

## 2.2 Neural network and function space

In this work, we consider a standard depth-$L$ width-$m$ fully connected ReLU neural network. Formally, we define a DNN with the output $\boldsymbol{s}_l(\boldsymbol{x})$ in each layer

$$\boldsymbol{s}_l(\boldsymbol{x}) = \begin{cases} \boldsymbol{x} & l = 0, \\ \phi(\langle \boldsymbol{W}_l, \boldsymbol{s}_{l-1}(\boldsymbol{x})\rangle) & 1 \leq l \leq L-1, \\ \langle \boldsymbol{W}_L, \boldsymbol{s}_{L-1}(\boldsymbol{x})\rangle & l = L, \end{cases} \tag{3}$$

where the input is $\boldsymbol{x} \in \mathbb{R}^d$, the output is $\boldsymbol{s}_L(\boldsymbol{x}) \in \mathbb{R}^d$, the weights of the neural networks are $\boldsymbol{W}_1 \in \mathbb{R}^{m\times d}$, $\boldsymbol{W}_l \in \mathbb{R}^{m\times m}$, $l = 2, \ldots, L-1$ and $\boldsymbol{W}_L \in \mathbb{R}^{d\times m}$. The neural network parameters formulate the tuple of weight matrices $\boldsymbol{W} := \{\boldsymbol{W}_i\}_{i=1}^L \in \{\mathbb{R}^{m\times d} \times (\mathbb{R}^{m\times m})^{L-2} \times \mathbb{R}^{d\times m}\}$. The $\mathcal{S}$ denotes the function space of Eq. (3).

The $\phi = \max(0, x)$ is the ReLU activation function. According to the property $\phi(x) = x\phi'(x)$ of ReLU, we have $\boldsymbol{s}_l = \boldsymbol{D}_l \boldsymbol{W}_l \boldsymbol{s}_{l-1}$, where $\boldsymbol{D}_l$ is a diagonal matrix defined as below.

**Definition 1** (Diagonal sign matrix). *For $l \in [L-1]$ and $k \in [m]$, the diagonal sign matrix $\boldsymbol{D}_l$ is defined as: $(\boldsymbol{D}_l)_{k,k} = 1\{(\boldsymbol{W}_l \boldsymbol{s}_{l-1})_k \geq 0\}$.*

**Initialization:** We make the standard random Gaussian initialization $[\boldsymbol{W}_l]_{i,j} \sim \mathcal{N}(0, \frac{2}{m})$ for $l \in [L-1]$ and $[\boldsymbol{W}_L]_{i,j} \sim \mathcal{N}(0, \frac{1}{d})$.

## 2.3 Causal discovery

In this paper, we follow the setting in Rolland et al. [2022] and consider the following causal model, a random variable $\boldsymbol{x} \in \mathbb{R}^d$ is generated by:

$$x^{(i)} = f_i(\text{PA}_i(\boldsymbol{x})) + \epsilon_i, \quad i \in [d], \tag{4}$$

where $f_i$ is a non-linear function, $\epsilon_i \sim \mathcal{N}(0, \sigma_i^2)$ and $\mathrm{PA}_i(\boldsymbol{x})$ represent the set of parents of $x^{(i)}$ in $\boldsymbol{x}$. Then we can write the probability distribution function of $\boldsymbol{x}$ as:

$$p(\boldsymbol{x}) = \prod_{i=1}^{d} p(x^{(i)} | \mathrm{PA}_i(\boldsymbol{x})) \,. \tag{5}$$

For such non-linear additive Gaussian noise models Eq. (4), Rolland et al. [2022] provides Algorithm 1 to learn the topological order by score matching as follows:

---

**Algorithm 1** SCORE matching causal order search (Adapted from Algorithm 1 in Rolland et al. [2022])

---

**Input:** training data $\{(\boldsymbol{x}_{(i)})_{i=1}^{N}\}$.
**Initialize:** $\pi = []$, nodes $= \{1, \dots, d\}$
**for** $k = 1, \dots, d$ **do**
    Estimate the score function $s_{\mathrm{nodes}} = \nabla \log p_{\mathrm{nodes}}$ by deep ReLU network with SGD.
    Estimate $V_j = \hat{\mathrm{Var}}_{\boldsymbol{x}_{\mathrm{nodes}}} \left[ \frac{\partial \boldsymbol{s}_j(\boldsymbol{x})}{\partial \boldsymbol{x}^{(j)}} \right]$.
    $l \leftarrow \mathrm{nodes}[\arg\min_j V_j]$
    $\pi \leftarrow [l, \pi]$
    nodes $\leftarrow$ nodes $- \{l\}$
    Remove $l$-th element of $\boldsymbol{x}$
**end for**
Get the final DAG by pruning the full DAG associated with the topological order $\pi$.

---

### 2.4 Score-based generative modeling (SGM)

In this section, we give a brief overview of SGM following Song et al. [2021], Chen et al. [2023b].

#### 2.4.1 Score-based generative modeling with SDEs

**Forward process:** The success of previous score-based generative modeling methods relies on perturbing data using multiple noise scales, and the proposal of the diffusion model is to expand upon this concept by incorporating an infinite number of noise scales. This will result in the evolution of perturbed data distributions as the noise intensity increases, which will be modeled through a stochastic differential equation (SDE).

$$\mathrm{d}\boldsymbol{x}_t = \boldsymbol{f}(\boldsymbol{x}_t, t)\mathrm{d}t + g_t \mathrm{d}\boldsymbol{w}, \quad \boldsymbol{x}_0 \sim p_0 \,. \tag{6}$$

The expression describes $\boldsymbol{x}_t$, where the standard Wiener process (also known as Brownian motion) is denoted as $\boldsymbol{w}$, the drift coefficient of $\boldsymbol{x}_t$ is represented by a vector-valued function called $\boldsymbol{f}$, and the diffusion coefficient of $\boldsymbol{x}_t$ is denoted as $g_t$, a scalar function. In this context, we will refer to the probability density of $\boldsymbol{x}_t$ as $p_t$, and the transition kernel from $\boldsymbol{x}_s$ to $\boldsymbol{x}_t$ as $p_{st}(\boldsymbol{x}_t|\boldsymbol{x}_s)$, where $0 \leq s < t \leq T$. The Ornstein–Uhlenbeck (OU) process is a Gaussian process that is both time-homogeneous and a Markov process. It is distinct in that its stationary distribution is equivalent to the standard Gaussian distribution $\gamma^d$ on $\mathbb{R}^d$.

**Reverse process:** We can obtain samples of $\boldsymbol{x}_0 \sim p_0^{\mathrm{SDE}}$ by reversing the process starting from samples of $\boldsymbol{x}_T \sim p_T^{\mathrm{SDE}}$. An important finding is that the reversal of a diffusion process is a diffusion process as well. It operates in reverse time and is described by the reverse-time SDE:

$$\mathrm{d}\boldsymbol{x}_t = \left( \boldsymbol{f}(\boldsymbol{x}_t, t) - g_t^2 \nabla_{\boldsymbol{x}} \log p_t(\boldsymbol{x}_t) \right) \mathrm{d}t + g_t \mathrm{d}\overline{\boldsymbol{w}} \,. \tag{7}$$

When time is reversed from $T$ to $0$, $\overline{\boldsymbol{w}}$ is a standard Wiener process with an infinitesimal negative timestep of $\mathrm{d}t$. The reverse diffusion process can be derived from Eq. (7) once the score of each marginal distribution, $\nabla \log p_t(\boldsymbol{x}_t)$, is known for all $t$. By simulating the reverse diffusion process, we can obtain samples from $p_0^{\mathrm{SDE}}$.

**Some special settings:** In order to simplify the writing of symbols and proofs, in this work we choose that $\boldsymbol{f}(\boldsymbol{x}_t, t) = -\frac{1}{2}\boldsymbol{x}_t$ and $g(t) = 1$ which has been widely employed in prior research [Chen et al., 2023a,b, De Bortoli et al., 2021] for theoretical analysis in Ornstein–Uhlenbeck process in score-based generative modeling.

### 2.4.2 Score matching in diffusion model

We aim to minimize the equivalent objective for score matching:

$$\min_{\boldsymbol{s}\in\mathcal{S}}\int_0^T w(t)\mathbb{E}_{\boldsymbol{x}_0\sim p_0}\left[\mathbb{E}_{\boldsymbol{x}_t\sim p_{0t}(\boldsymbol{x}_t|\boldsymbol{x}_0)}\left[\left\|\nabla_{\boldsymbol{x}_t}\log p_{0t}(\boldsymbol{x}_t|\boldsymbol{x}_0)-\boldsymbol{s}(\boldsymbol{x}_t,t)\right\|_2^2\right]\right]\mathrm{d}t\,.$$

The transition kernel has an analytical form $\nabla_{\boldsymbol{x}_t}\log p_{0t}(\boldsymbol{x}_t|\boldsymbol{x}_0)=-\frac{\boldsymbol{x}_t-\alpha(t)\boldsymbol{x}_0}{h(t)}$, where $\alpha(t)=e^{-\frac{t}{2}}$ and $h(t)=1-\alpha(t)^2=1-e^{-t}$.

The empirical score matching loss is:

$$\min_{\boldsymbol{s}\in\mathcal{S}}\hat{\mathcal{L}}(\boldsymbol{s})=\frac{1}{n}\sum_{i=1}^n\ell(\boldsymbol{x}_{(i)};\boldsymbol{s})\,, \tag{8}$$

where the loss function $\ell(\boldsymbol{x}_{(i)};\boldsymbol{s})$ is defined as:

$$\ell(\boldsymbol{x}_{(i)};\boldsymbol{s})=\frac{1}{T-t_0}\int_{t_0}^T\mathbb{E}_{\boldsymbol{x}_t\sim p_{0t}(\boldsymbol{x}_t|\boldsymbol{x}_0=\boldsymbol{x}_{(i)})}\left[\left\|\nabla_{\boldsymbol{x}_t}\log p_{0t}(\boldsymbol{x}_t|\boldsymbol{x}_0=\boldsymbol{x}_{(i)})-\boldsymbol{s}(\boldsymbol{x}_t,t)\right\|_2^2\right]\mathrm{d}t\,.$$

Here we choose $w(t)=\frac{1}{T-t_0}$, and we define the expected loss $\mathcal{L}(\cdot)=\mathbb{E}_{\boldsymbol{x}\sim p_0}[\hat{\mathcal{L}}(\cdot)]$.

## 3 Theoretical results for causal discovery

In this section, we state the main theoretical results of this work. We present the assumptions on non-linear additive Gaussian noise causal models in Section 3.1. Then, we present the sample complexity bound for score matching in causal discovery in Section 3.2. In Section 3.3 we provide the upper bound on the error rate for causal discovery using the Algorithm 1. The full proofs of Theorem 1 and 2 are deferred to Appendix E and F, respectively.

### 3.1 Assumptions

**Assumption 1** (Lipschitz property of score function). *The score function $\nabla\log p(\cdot)$ is 1-Lipschitz.*

**Remark:** The Lipschitz property of the score function is a standard assumption commonly used in the existing literature [Block et al., 2020, Lee et al., 2022, Chen et al., 2023b,a]. However, for causal discovery, this assumption limits the family of mechanisms that we can cover.

**Assumption 2** (Structural assumptions of causal model). *Let $p$ be the probability density function of a random variable $\boldsymbol{x}$ defined via a non-linear additive Gaussian noise model Eq. (4). Then, $\forall i\in[d]$ the non-linear function is bounded, $|f_i|\leq C_i$. And $\forall i,j\in[d]$, if $j$ is one of the parents of $i$, i.e. $x^{(j)}\Rightarrow x^{(i)}$, then there exist a constant $C_m$ that satisfy:*

$$\mathbb{E}_{p(\boldsymbol{x})}\left(\frac{\partial^2 f_i(PA_i(\boldsymbol{x}))}{\partial x^{(j)2}}\right)^2\geq C_m\sigma_i^2\,.$$

**Remark:** This is a novel assumption that we introduce, relating the average second derivative of a mechanism (related to its curvature) to the noise variance of the child variable. This will play a crucial yet intuitive role in our error bound: identifiability is easier when there is sufficient non-linearity of a mechanism with respect to the noise of the child variable. Consider the example of a quadratic mechanism, where the second derivative is the leading constant of the polynomial. If this constant is small (e.g., close to zero), the mechanism is almost linear and we may expect that the causal model should be harder to identify. Similarly, if the child variable has a very large variance, one may expect it to be more difficult to distinguish cause from effect, as the causal effect of the parent is small compared to the noise of the child. According to Assumption 2, we can derive the identified ability margin for leaf nodes and parent nodes.

**Lemma 1.** *If a non-linear additive Gaussian noise model Eq. (4) satisfies Assumption 2. Then, $\forall i,j\in[d]$, we have:*

$$i\text{ is a leaf}\Rightarrow Var\left(\frac{\partial s_i(\boldsymbol{x})}{\partial x^{(i)}}\right)=0,\ j\text{ is not a leaf}\Rightarrow Var\left(\frac{\partial s_j(\boldsymbol{x})}{\partial x^{(j)}}\right)\geq C_m.$$

This lemma intuitively relates our identifiability margin with the decision rule of SCORE Rolland et al. [2022] to identify leaves. Non-leaf nodes should have the variance of their score Jacobian sufficiently far from zero. As one may expect, we will see in Theorem 2 that the closer $C_m$ is to zero, the more likely it is that the result of the algorithm will be incorrect given finite samples.

## 3.2 Error bound for score matching in causal discovery

We are now ready to state the main result of the score matching in causal discovery. We provide the sample complexity bounds of the explicit score matching Eq. (1) that using denoising score matching Eq. (2) in Algorithm 1 for non-linear additive Gaussian noise models Eq. (4).

**Theorem 1.** *Given a DNN defined by Eq. (3) trained by SGD for minimizing empirical denoising score matching objective. Suppose Assumption 1 and 2 are satisfied. For any $\varepsilon \in (0, 1)$ and $\delta \in (0, 1)$, if $\sigma_i \asymp \sigma$ and $\frac{C_i}{\sigma_i} \asymp 1$, $\forall i \in [d]$. Then with probability at least $1 - 2\delta - 4\exp(-\frac{d}{32}) - 2L\exp(-\Omega(m)) - \frac{1}{nd}$ over the randomness of initialization $\boldsymbol{W}$, noise $\boldsymbol{\epsilon}$ and $\epsilon_i$, it holds that:*

$$J_{ESM}(\hat{\boldsymbol{s}}, p(\boldsymbol{x})) \lesssim \frac{\sigma^2 d \log nd}{n\varepsilon^2} \log \frac{\mathcal{N}_c(\frac{1}{n}, \mathcal{S})}{\delta} + \frac{1}{n} + d\varepsilon^2 \,,$$

*where the $\mathcal{N}_c(\frac{1}{n}, \mathcal{S})$ is the covering number of the function space $\mathcal{S}$ for deep ReLU neural network.*

**Remark:**

**1):** To the best of our knowledge, our results present the first upper bound on the explicit sampling complexity of score matching for topological ordering Algorithm 1 in non-linear additive Gaussian noise causal models. This novel contribution provides valuable insights into the efficiency and effectiveness of utilizing score matching for topological ordering in non-linear additive Gaussian noise causal models.

**2):** By choosing $\varepsilon^2 = \frac{1}{\sqrt{n}}$, the bound is modified to $J_{\text{ESM}}(\hat{\boldsymbol{s}}, p(\boldsymbol{x})) \lesssim \frac{\sigma^2 d \log nd}{\sqrt{n}} \log \frac{\mathcal{N}_c(1/n, \mathcal{S})}{\delta}$. This expression demonstrates that the $\ell_2$ estimation error converges at a rate of $\frac{\log n}{\sqrt{n}}$ when the sample size $n$ is significantly larger than the number of nodes $d$.

**3):** The bound is also related to the number of nodes $d$, the variance of the noise in denoising score matching $\sigma$ and causal model $\sigma_i$, the covering number of the function space $\mathcal{N}_c(\frac{1}{n}, \mathcal{S})$, and the upper bound of the data $C_d$. If these quantities increase, it is expected that the error of explicit score matching will also increase. This is due to the increased difficulty in accurately estimating the score function.

**4):** Theorem 1 is rooted in the generalization by sampling complexity bound. It is independent of the specific training algorithm used. The results are broadly applicable and can be seamlessly extended to encompass larger batch GD.

Next, we will establish a connection between score matching and the precise identification of the topological ordering.

## 3.3 Error bound for topological order in causal discovery

Based on the previously mentioned sample complexity bound of score matching, we establish an upper bound on the error rate of the topological ordering of the causal model obtained through Algorithm 1.

**Theorem 2.** *Given a DNN defined by Eq. (3) trained by SGD with a step size $\eta = \mathcal{O}(\frac{1}{poly(n,L)m\log^2 m})$ for minimizing empirical score matching objective. Then under Assumption 2, for $m \geq poly(n, L)$, with probability at least:*

$$1 - \exp(-\Theta(d)) - (L+1)\exp(-\Theta(m)) - 2n\exp(-\frac{nC_m^2 d^2}{2^{4L+5}(\log m)^2(m^2 + d^2)}) \,,$$

*over the randomness of initialization $\boldsymbol{W}$ and training data that Algorithm 1 can completely recover the correct topological order of the non-linear additive Gaussian noise model.*

**Remark:**

**1):** The foundation of Theorem 2 rests upon Theorem 1, it can be seen as an embodiment of applying the upper bound of score matching for causal discovery. To the best of our knowledge, our results provide the first upper bound on the error rate of topological ordering in non-linear additive Gaussian noise causal models using Algorithm 1.

**2):** Considering that when $m \asymp d$ and $L \asymp 1$ the probability degenerates to:

$$1 - \Theta(e^{-m}) - 2n \exp\left(-\Theta\left(\frac{nC_m^2}{(\log m)^2}\right)\right).$$

The first term of the error arises due to the initialization of the neural network. As for the second term of the error, if the number of training data $n$ satisfies $\frac{n}{\log n} \gtrsim (\log m)^2$, then it will have that $2n \exp\left(-\Theta\left(\frac{nC_m^2}{(\log m)^2}\right)\right) \lesssim 1$. This implies that the second term of the error probability exhibits linear convergence towards 0 when $n$ is sufficiently large. Therefore, when the sample size $\frac{n}{\log n} \gtrsim (\log m)^2$, the contribution of the second term to the full error becomes negligible.

**3):** The theorem reveals that a smaller value of the constant $C_m$ increases the probability of algorithm failure. This observation further confirms our previous statement that a smaller average second derivative of the nonlinear function makes it more challenging to identify the causal relationship in the model. Additionally, when the causal relationship is linear, our theorem does not provide any guarantee for the performance of Algorithm 1.

**4):** Consider the two variables case. If a child node is almost a deterministic function of its parents, the constant $C_m$ can take on arbitrarily large values, according to Assumption 2. Consequently, the second term of the error probability, $2n \exp\left(-\Theta\left(\frac{nC_m^2}{(\log m)^2}\right)\right)$, tends to zero. This implies that the errors in Algorithm 1 are primarily caused by the random initialization of the neural network. The identifiability of this setting is consistent with classical results Daniušis et al. [2010], Janzing et al. [2015]. Intuitively, as long as the non-linearity is chosen independently of the noise of the parent variable[2], the application of the non-linearity will increase the distance to the reference distribution of the parent variable (in our case Gaussian). Note that for the derivative in Assumption Assumption 2 to be defined, the parent node cannot be fully deterministic.

**5):** Instead of focusing on the kernel regime, we directly cover the more general neural network training. The kernel approach of Rolland et al. [2022] is a special case of our analysis. The basis of Theorem 2 lies in the proof of SGD/GD convergence of the neural network, These convergence outcomes also apply to BatchGD, as demonstrated in Jentzen and Kröger [2021]. Hence, Theorem 2 can naturally be expanded to incorporate Batch GD as well.

**Proof sketch:** The proof of Theorem 2 can be divided into three steps. The first and most important step is to derive the upper bound of $\frac{\partial s_i(\boldsymbol{x})}{\partial x^{(i)}}$. Here, we utilize the properties of deep ReLU neural networks to derive the distribution relationship between features of adjacent layers, then accumulate them and combine it with the properties of Gaussian initialization, yielding the upper bound for $\frac{\partial s_i(\boldsymbol{x})}{\partial x^{(i)}}$. The second step is to use the upper bound of $\frac{\partial s_i(\boldsymbol{x})}{\partial x^{(i)}}$ obtained in the first step combined with the concentration inequality to derive the upper bound of the error of $\mathrm{Var}\left(\frac{\partial s_i(\boldsymbol{x})}{\partial x^{(i)}}\right)$. The third step is to compare the upper bound in the second step with Lemma 1 to obtain the probability of successfully selecting leaf nodes in each step. After accumulation, we can obtain the probability that Algorithm 1 can completely recover the correct topological order of the non-linear additive Gaussian noise model.

## 4  Theoretical results for score-based generative modeling (SGM)

In this section, we present the additional assumption required for the theoretical analysis of score matching in score-based generative modeling. Then, we provide the sample complexity bound associated with score matching in this framework. The full proof in this section is deferred to Appendix G.

**Assumption 3** (Bounded data). *We assume that the input data satisfy* $\|\boldsymbol{x}\|_2 \leq C_d$, $\boldsymbol{x} \sim p_0$.

**Remark:** Bounded data is standard in deep learning theory and also commonly used in practice [Du et al., 2019b,a, Allen-Zhu et al., 2019, Oymak and Soltanolkotabi, 2020, Malach et al., 2020].

---

[2] Daniušis et al. [2010], Janzing et al. [2015] have formalized independence of distribution and function via an information geometric orthogonality condition that refers to a reference distribution (e.g., Gaussian)

**Theorem 3.** *Given a DNN defined by Eq. (3) trained by SGD for minimizing empirical denoising score matching loss Eq. (8). Suppose Assumption 1 and 3 are satisfied. For any $\varepsilon \in (0,1)$ and $\delta \in (0,1)$. Then with probability at least $1 - 2\delta - 2L\exp(-\Omega(m))$ over the randomness of initialization $\mathbf{W}$ and noise $\boldsymbol{\epsilon}$ in denoising score matching, it holds:*

$$\frac{1}{T-t_0}\int_{t_0}^{T}\|\nabla\log p_t(\cdot) - \hat{\boldsymbol{s}}(\cdot,t)\|_{\ell^2(p_t)}^2\,\mathrm{d}t \lesssim \frac{1}{n\varepsilon^2}\left(\frac{d(T-\log(t_0))}{T-t_0} + C_d^2\right)\log\frac{\mathcal{N}_c(\frac{1}{n},\mathcal{S})}{\delta} + \frac{1}{n} + d\varepsilon^2,$$

*where the $\mathcal{N}_c(\frac{1}{n},\mathcal{S})$ is the covering number of the function space $\mathcal{S}$ for deep ReLU neural network.*

**Remark:**

**1):** Theorem 3 and Theorem 1 study similar problems between causal discovery and score-based generative modeling and share similar techniques drawn from statistical learning theory and deep learning theory. These two domains are connected by a common theoretical foundation centered on the upper bound of score matching.

**2):** Our result extends the results for score matching in diffusion models presented in Chen et al. [2023a] which rested on the assumption of low-dimensional data structures, employing this to decompose the score function and engineer specialized network architectures for the derivation of the upper bound. Our work takes a distinct route. Our conclusions are based on the general deep ReLU neural network instead of a specific encoder-decoder network and do not rely on the assumptions of low-dimensional data used in Chen et al. [2023a]. We harness the inherent traits and conventional techniques of standard deep ReLU networks to directly deduce the upper error bound. This broader scope allows for a more comprehensive understanding of the implications and applicability of score-based generative modeling in a wider range of scenarios.

**3):** Similar to Theorem 1, by choosing $\varepsilon^2 = \frac{1}{\sqrt{n}}$, we can obtain the best bound $\frac{1}{T-t_0}\int_{t_0}^{T}\|\nabla\log p_t(\cdot) - \hat{\boldsymbol{s}}(\cdot,t)\|_{\ell^2(p_t)}^2\,\mathrm{d}t \lesssim \frac{1}{\sqrt{n}}\left(\frac{d(T-\log(t_0))}{T-t_0} + C_d^2\right)\log\frac{\mathcal{N}_c(\frac{1}{n},\mathcal{S})}{\delta}$. This expression demonstrates that the $\ell_2$ estimation error converges at a rate of $\frac{1}{\sqrt{n}}$ when the sample size $n$ is significantly larger than the dimensionality $d$ and time steps $T$.

**4):** The bound is also related to the data dimension $d$, the variance of the noise in denoising score matching $\sigma$, the covering number of the function space $\mathcal{N}_c(\frac{1}{n},\mathcal{S})$, and the upper bound of the data $C_d$. If these quantities increase, it is expected that the error of explicit score matching will also increase. This is due to the increased difficulty in accurately estimating the score function.

**5):** When $t_0 = 0$, the theorem lacks meaning. However, when $T \gg t_0 \approx 1$, the bound simplifies to $\frac{d+C_d^2}{\sqrt{n}}\log\frac{\mathcal{N}_c(\frac{1}{n},\mathcal{S})}{\delta}$. This indicates that when $T$ is sufficiently large, the loss estimated by the score function in the diffusion model becomes independent of time steps $T$.

**6):** Similar to Theorem 1, the result of Theorem 3 is also broadly applicable and can be seamlessly extended to encompass larger batch GD.

## 5 Numerical evidence

We conducted a series of experiments to validate the theoretical findings presented in the paper. We took inspiration from the code provided inRolland et al. [2022] and employed the structural Hamming distance (SHD) between the generated output and the actual causal graph to assess the outcomes. The ensuing experimental outcomes for SHD, vary across causal model sizes $d$, sample sizes $n$, and $C_m$. The experimental results are shown in Tables 1 to 3

Table 1: Fixed model size $d = 100$ and the number of sampling $n = 100$, SHD results of causal discovery using Algorithm 1 for different $C_m$ values (10 runs).

| $C_m$ | 1 | 2 | 4 | 8 | 16 |
|---|---|---|---|---|---|
| SHD | $2941.0 \pm 29.5$ | $2905.7 \pm 50.8$ | $2900.6 \pm 80.8$ | $2637.1 \pm 200.4$ | $1512.4 \pm 283.6$ |
| $C_m$ | 32 | 64 | 128 | 256 | 512 |
| SHD | $413.9 \pm 93.4$ | $55.0 \pm 16.0$ | $23.9 \pm 4.6$ | $21.2 \pm 5.0$ | $13.8 \pm 1.8$ |

Table 2: Fixed model size $d = 10$ and $C_m = 1$, SHD results of causal discovery using Algorithm 1 for the different number of sampling $n$ (10 runs).

| $n$ | 5 | 10 | 20 | 40 | 80 | 100 | 160 |
|---|---|---|---|---|---|---|---|
| SHD | $31.7 \pm 2.1$ | $27.8 \pm 4.1$ | $23.3 \pm 2.7$ | $23.0 \pm 4.0$ | $18.4 \pm 3.3$ | $16.5 \pm 3.4$ | $13.0 \pm 4.0$ |

Table 3: Fixed the number of sampling $n = 10$ and $C_m = 1$, SHD results of causal discovery using Algorithm 1 for the different model size $d$ (10 runs).

| $d$ | 5 | 10 | 20 | 40 | 80 | 100 |
|---|---|---|---|---|---|---|
| SHD | $4.5 \pm 2.0$ | $29.6 \pm 2.2$ | $124.3 \pm 4.6$ | $522.8 \pm 11.6$ | $1965.4 \pm 18.7$ | $2923.7 \pm 38.5$ |

Analyzing the experimental outcomes, we find a notable pattern: higher values of $C_m$, augmented sample sizes $n$, and reduced model size $d$ all contribute to the performance of Algorithm 1 which is consistent with the insights from Theorem 2.

## 6  Related Work

**Score matching:**   Score Matching was initially introduced by Hyvärinen [2005] and extended to energy-based models by Song and Ermon [2019]. Subsequently, Vincent [2011] proposed denoising score matching, which transforms the estimation of the score function for the original distribution into an estimation for the noise distribution, effectively avoiding the need for second derivative computations. Other methods, such as sliced score matching [Song et al., 2020], denoising likelihood score matching [Chao et al., 2022], and kernel-based estimators, have also been proposed for score matching. The relationship between score matching and Fisher information [Shao et al., 2019], as well as Langevin dynamics [Hyvarinen, 2007], has been explored. On the theoretical side, Wenliang and Kanagawa [2020] introduced the concept of "blindness" in score matching, while Koehler et al. [2023] compared the efficiency of maximum likelihood and score matching, although their results primarily focus on exponential family distributions. Our paper, for the first time, analyzes the sample complexity bounds of the score function estimating in causal inference.

**Causal discovery:**   The application of score methods for causal inference for linear additive models began with Ghoshal and Honorio [2018], which proposed a causal structure recovery method based on topological ordering from the precision matrix (equivalent to the score in that setting). Under certain noise variance assumptions, their method can reliably recover the DAG in polynomial time and sample complexity.

In recent years, there have been numerous algorithms developed for causal inference in non-linear additive models. GraNDAG [Lachapelle et al., 2021] aims to maximize the likelihood of the observed data under this model and enforces a continuous constraint to ensure the acyclicity of the causal graph Rolland et al. [2022] proposed a novel approach for causal inference which utilize score matching algorithms as a foundation for topological ordering and then employ sparse regression techniques to prune the DAG. Subsequently, Montagna et al. [2023a] extended the method to non-Gaussian noise, Sanchez et al. [2023] proposed to use diffusion models to fit the score function, and Montagna et al. [2023b] proposed a new scalable score-based preliminary neighbor search techniques.

Although advances have been achieved in leveraging machine learning for causal discovery, there is generally a lack of further research on error bounds. Other studies concentrate on broader non-parametric models but depend on various assumptions like faithfulness, restricted faithfulness, or the sparsest Markov representation [Spirtes et al., 2000, Raskutti and Uhler, 2018, Solus et al., 2021]. These approaches employ conditional independence tests and construct a graph that aligns with the identified conditional independence relations [Zhang, 2008].

**Theoretical analysis of score-based generative modeling:**   Existing work mainly focuses on two fundamental questions: "How do diffusion models utilize the learned score functions to estimate the data distribution?" [Chen et al., 2023b, De Bortoli et al., 2021, De Bortoli, 2022, Lee et al., 2022,

2023] and "Can neural networks effectively approximate and learn score functions? What are the convergence rate and bounds on the sample complexity?" [Chen et al., 2023a].

Specifically, De Bortoli et al. [2021] and Lee et al. [2022] studied the convergence guarantees of diffusion models under the assumptions that the score estimator is accurate under the $\ell_1$ and $\ell_2$ norms. Concurrently Chen et al. [2023b] and Lee et al. [2023] extended previous results to distributions with bounded moments. De Bortoli [2022] studied the distribution estimation guarantees of diffusion models for low-dimensional manifold data under the assumption that the score estimator is accurate under the $\ell_1$ or $\ell_2$ norms.

However, these theoretical results rely on the assumption that the score function is accurately estimated, while the estimation of the score function is largely untouched due to the non-convex training dynamics. Recently, Chen et al. [2023a] provided the first sample complexity bounds for score function estimation in diffusion models. However, their result is based on the assumption that the data distribution is supported on a low-dimensional linear subspace and they use a specialized Encoder-Decoder network instead of a general deep neural network. As a result, a complete theoretical picture of score-based generative modeling is still lacking.

# 7  Conclusion and Limitations

In this work, we investigate the sample complexity error bounds of Score Matching using deep ReLU neural networks under two different problem settings: causal discovery and score-based generative modeling. We provide a sample complexity analysis for the estimation of the score function in the context of causal discovery for nonlinear additive Gaussian noise models, with a convergence rate of $\frac{\log n}{n}$. Furthermore, we extend the sample complexity bounds for the estimation of the score function in the ScoreSDE method to general data and achieve a convergence rate of $\frac{1}{n}$. Additionally, we provide an upper bound on the error rate of the state-of-the-art causal discovery method SCORE [Rolland et al., 2022], showing that the error rate of this algorithm converges linearly with respect to the number of training data.

A core limitation of this work is limiting our results to the Gaussian noise assumption. In fact, non-linear mechanisms with additive non-gaussian noise are also identifiable under mild additional assumptions [Peters et al., 2014] and Montagna et al. [2023a] already extended the score-matching approach of Rolland et al. [2022] to that setting. Relaxing this assumption would also allow us to apply our bounds to interesting corner cases, such as linear non-gaussian [Ghoshal and Honorio, 2018], and non-gaussian deterministic causal relations [Daniušis et al., 2010, Janzing et al., 2015]. It may be possible for this assumption to be relaxed in future work, but we argue that the added challenge, the significant difference in algorithms, and the standalone importance of the non-linear Gaussian case justify our focus.

In addition, we make other assumptions that limit the general applicability of our bounds. In particular, the assumption of the Lipschitz property for the score function imposes a strong constraint on the model space. Further investigating the relationship between the noise, the properties of the nonlinear functions in the causal model Eq. (4), and the resulting Lipschitz continuity of the score function would be an interesting extension of this work.

## Acknowledgements

We are thankful to the reviewers for providing constructive feedback and Kun Zhang and Dominik Janzing for helpful discussion on the special case of deterministic children. This work was supported by Hasler Foundation Program: Hasler Responsible AI (project number 21043). This work was supported by the Swiss National Science Foundation (SNSF) under grant number 200021_205011. Francesco Locatello did not contribute to this work at Amazon. Corresponding author: Zhenyu Zhu.

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
