## Appendix introduction

The Appendix is organized as follows:

- In Appendix A, we provide a summary of the symbols and notations used throughout this paper.

- In Appendix B, we provide some background to some of the content covered in this paper.

- In Appendix C, we present several relevant lemmas that are essential to the proofs in this paper.

- In Appendix D, we provide the proof of Lemma 1.

- In Appendix E, we provide the proof of Theorem 1.

- In Appendix F, we provide the proof of Theorem 2.

- In Appendix G, we provide the proof of Theorem 3.

- In Appendix H, we discuss the Assumption 1, the Lipschitz property of score function.

- Finally, in Appendix I, we discuss the broader impacts of this paper.

## A    Symbols and Notation

In the paper, vectors are indicated with bold small letters, and matrices with bold capital letters. To facilitate the understanding of our work, we include some core symbols and notation in Table 4.

Table 4: Core symbols and notations used in this project.

| Symbol | Dimension(s) | Definition |
|---|---|---|
| $\mathcal{S}$ | - | Function space |
| $\mathcal{N}_c(\cdot, \mathcal{S})$ | $\mathbb{R}$ | Covering number of function space $\mathcal{S}$ |
| $\mathcal{N}(\mu, \sigma^2)$ | - | Gaussian distribution with mean $\mu$ and variance $\sigma^2$ |
| $p$ | - | Probability density function of a probability distribution |
| $\mathbb{E}$ | - | Expected value |
| $[L]$ | - | Shorthand of $\{1, 2, \ldots, L\}$ |
| $\mathcal{O}, o, \Omega$ and $\Theta$ | - | Standard Bachmann–Landau order notation |
| $n$ | $\mathbb{R}$ | Number of data |
| $d$ | $\mathbb{R}$ | Data dimension (number of variables in the causal model) |
| $L$ | $\mathbb{R}$ | Depth of the neural network |
| $m$ | $\mathbb{R}$ | Width of the neural network |
| $\phi$ | - | The ReLU activation function |
| $x^{(i)}$ | $\mathbb{R}$ | The $i$-th element of the vector $\boldsymbol{x}$ |
| $\boldsymbol{x}_{(i)}$ | $\mathbb{R}^d$ | The $i$-th data point |
| $\boldsymbol{x}_t$ | $\mathbb{R}^d$ | The data point in time $t$ in diffusion model |
| $\boldsymbol{W}_1$ | $\mathbb{R}^{m \times d}$ | Weight matrix for the input layer |
| $\boldsymbol{W}_l$ | $\mathbb{R}^{m \times m}$ | Weight matrix for the $l$-th hidden layer |
| $\boldsymbol{W}_L$ | $\mathbb{R}^{d \times m}$ | Weight matrix for the output layer |
| $\epsilon$ | $\mathbb{R}$ | The noise introduced by denoising score matching |
| $\sigma$ | $\mathbb{R}$ | The standard deviation of Gaussian noise $\epsilon$ |
| $\epsilon_i$ | $\mathbb{R}$ | The noise of $i$-th variable of causal model |
| $\sigma_i$ | $\mathbb{R}$ | The standard deviation of Gaussian noise $\epsilon_i$ |
| $f_i$ | - | Non-linear function of $i$-th variable of causal model |
| $\text{PA}_i(\boldsymbol{x})$ | - | The set of parents of $x^{(i)}$ in $\boldsymbol{x}$ |
| $\text{CH}_j(\boldsymbol{x})$ | - | The set of children of $x^{(j)}$ in $\boldsymbol{x}$ |

# B  More backgrounds

## B.1  Covering number

The basic idea of covering number is to approximate a function space with an infinite number of elements by a finite number of elements. It is used to describe how many elements (or subsets) in a given metric space can be "covered" with a finite number of reference elements (or reference subsets) to ensure that the entire space is covered. It is defined as follows:

**Definition 2.** *We assume there exists $m = m(\epsilon)$ elements $f_1, \ldots, f_m$ such that for any $f \in \mathcal{F}, \exists i \in \{1, \ldots, m\}$ such that $d(f, f_i) \leq \epsilon$. The minimal possible number $m(\epsilon)$ is the covering number of $\mathcal{F}$ at precision $\epsilon$.*

In learning theory, covering number can be used to bound the Rademacher complexity [Shalev-Shwartz and Ben-David, 2014] then it is related to generalization.

## B.2  More backgrounds about Algorithm 1

The main source of inspiration of the Rolland et al. [2022] to design Algorithm 1 is the following lemma:

**Lemma 2** (Adapted from Lemma 1 in Rolland et al. [2022])**.** *Let $p$ be the probability density function of a random variable $\boldsymbol{x}$ defined via a non-linear additive Gaussian noise model Eq. (4), and let $\boldsymbol{s}(\boldsymbol{x}) = \nabla \log p(\boldsymbol{x})$ be the associated score function. Then, $\forall j \in [d]$, we have:*

1. *$j$ is a leaf $\Leftrightarrow \forall \boldsymbol{x}, \frac{\partial s_j(\boldsymbol{x})}{\partial x^{(j)}} = c$, with $c \in \mathbb{R}$ independent of $\boldsymbol{x}$, i.e., $Var\big(\frac{\partial s_j(\boldsymbol{x})}{\partial x^{(j)}}\big) = 0$.*

2. *$j$ is a leaf, $i$ is a parent of $j \Leftrightarrow s_j(\boldsymbol{x})$ depends on $\boldsymbol{x}^{(i)}$, i.e., $Var\big(\frac{\partial s_j(\boldsymbol{x})}{\partial x^{(i)}}\big) \neq 0$.*

Lemma 2 reveals the important properties of the nonlinear additive Gaussian noise model: for non-linear additive Gaussian noise models, leaf nodes (and only leaf nodes) have the property that the associated diagonal element in the score's Jacobian is a constant. Therefore, by repeating this method and always removing the identified leaves, we can estimate a full topological order. This procedure is summarized in Algorithm 1.

# C  Relevant Lemmas

**Lemma 3** (Adapted from Lemma 10 in Chen et al. [2020])**.** *For any $\varepsilon \in (0, 1)$ and any target 1-Lipschitz function $\tilde{\boldsymbol{s}}$ that defined on $[0, 1]^d$ with $\tilde{\boldsymbol{s}}(0) = 0$, the architecture yields an approximation $\boldsymbol{s} \in \mathcal{S}$ satisfying $\|\boldsymbol{s} - \tilde{\boldsymbol{s}}\|_\infty \leq \varepsilon$.*

*The configuration of network architecture is:*

$$
\begin{aligned}
\|\boldsymbol{s}_l\|_\infty &\leq \sqrt{d}, \quad l \in [L], \\
\|\boldsymbol{W}_l\|_\infty &\leq \mathcal{O}(1), \quad l \in [L], \\
L &= \mathcal{O}(\log \frac{1}{\varepsilon} + d), \\
m &= \mathcal{O}(\frac{1}{\varepsilon^d}), \\
\sum_{l=1}^{L} \|\boldsymbol{W}_l\|_0 &\leq \mathcal{O}(\frac{1}{\varepsilon^d}(\log \frac{1}{\varepsilon} + d)).
\end{aligned}
$$

**Lemma 4** (Adapted from Theorem 4.4.5 in Vershynin [2018])**.** *Let $\boldsymbol{W}$ be an $N \times n$ matrix whose entries are independent standard normal random variables. Then for every $t \geq 0$, with probability at least $1 - 2\exp(-t^2/2)$, one has:*

$$
s(\boldsymbol{A})_{\max} \leq \sqrt{N} + \sqrt{n} + t,
$$

*where the $s(\boldsymbol{W})_{\max}$ represent the largest singular value of $\boldsymbol{W}$.*

**Lemma 5.** *If a causal model Eq. (4) satisfies Assumption 2. Then with probability at least $1 - \frac{1}{n^2 d}$ we have:*

$$\|\boldsymbol{x}\|_2^2 \le \sum_{i=1}^d (C_i + 2\sigma_i \sqrt{\log nd})^2.$$

*Proof.* Firstly, we can derive that:

$$\|\boldsymbol{x}\|_2^2 = \sum_{i=1}^d (x^{(i)})^2 = \sum_{i=1}^d (f_i + \epsilon_i)^2 \le \sum_{i=1}^d (C_i + |\epsilon_i|)^2.$$

Since $\epsilon_i \sim \mathcal{N}(0, \sigma_i^2)$, according to the tail bound of Gaussian distribution, with probability at least $1 - \exp(-\frac{t_i^2}{2\sigma_i^2})$ we have $|\epsilon_i| \le t_i$. Thus:

$$\|\boldsymbol{x}\|_2^2 \le \sum_{i=1}^d (C_i + t_i)^2,$$

with probability at least $1 - \sum_{i=1}^d \exp(-\frac{t_i^2}{2\sigma_i^2})$.

Choose $t_i = 2\sigma_i \sqrt{\log nd}$, then we have:

$$\|\boldsymbol{x}\|_2^2 \le \sum_{i=1}^d (C_i + 2\sigma_i \sqrt{\log nd})^2,$$

with probability at least $1 - \frac{1}{n^2 d}$.

$\square$

## D  Proof of Lemma 1

*Proof.* According to Eq. (5), we can derive that:

$$
\begin{aligned}
\log p(\boldsymbol{x}) &= \sum_{i=1}^d \log p(x^{(i)}|\mathrm{PA}_i(\boldsymbol{x})) \\
&= -\frac{1}{2} \sum_{i=1}^d \left( \frac{x^{(i)} - f_i(\mathrm{PA}_i(\boldsymbol{x}))}{\sigma_i} \right)^2 - \frac{1}{2} \sum_{i=1}^d \log(2\pi\sigma_i^2).
\end{aligned}
$$

Then:

$$s_j(\boldsymbol{x}) = \frac{f_j(\mathrm{PA}_j(\boldsymbol{x})) - x^{(j)}}{\sigma_j^2} + \sum_{i \in \mathrm{CH}_j(\boldsymbol{x})} \frac{\partial f_i(\mathrm{PA}_i(\boldsymbol{x}))}{\partial x^{(j)}} \frac{\epsilon_i}{\sigma_i^2}. \tag{9}$$

If $j$ is a leaf:

$$\frac{\partial s_j(\boldsymbol{x})}{\partial x^{(j)}} = -\frac{1}{\sigma_j^2}, \quad \mathrm{Var}\left( \frac{\partial s_j(\boldsymbol{x})}{\partial x^{(j)}} \right) = 0. \tag{10}$$

If $j$ is not a leaf:

$$\frac{\partial s_j(\boldsymbol{x})}{\partial x^{(j)}} = -\frac{1}{\sigma_j^2} + \sum_{i \in \mathrm{CH}_j(\boldsymbol{x})} \frac{\partial^2 f_i(\mathrm{PA}_i(\boldsymbol{x}))}{\partial x^{(j)2}} \frac{\epsilon_i}{\sigma_i^2},$$

where the $\mathrm{PA}_i(\boldsymbol{x})$ represent the set of parents of $x^{(i)}$ in $\boldsymbol{x}$. Then, according to the independence of $\epsilon_i$:

$$
\begin{aligned}
\mathrm{Var}\left(\frac{\partial s_j(\boldsymbol{x})}{\partial x^{(j)}}\right) &= \sum_{i \in \mathrm{CH}_j(\boldsymbol{x})} \mathrm{Var}\left(\frac{\partial^2 f_i(\mathrm{PA}_i(\boldsymbol{x}))}{\partial x^{(j)2}} \frac{\epsilon_i}{\sigma_i^2}\right) \\
&\geq \mathrm{Var}\left(\frac{\partial^2 f_i(\mathrm{PA}_i(\boldsymbol{x}))}{\partial x^{(j)2}} \frac{\epsilon_i}{\sigma_i^2}\right) \quad \forall i \in \mathrm{CH}_j(\boldsymbol{x}) \\
&= \mathbb{E}_{p(\boldsymbol{x})}\left(\frac{\partial^2 f_i(\mathrm{PA}_i(\boldsymbol{x}))}{\partial x^{(j)2}}^2\right) \mathrm{Var}\left(\frac{\epsilon_i}{\sigma_i^2}\right) \quad \forall i \in \mathrm{CH}_j(\boldsymbol{x}) \\
&\geq C_m .
\end{aligned}
\tag{11}
$$

Combine Eqs. (10) and (11), which concludes the proof. $\qquad\square$

# E Proof of the error bound of score function estimate for the causal model (Theorem 1)

*Proof.* Firstly, we use oracle inequality to decompose $J_{\mathrm{DSM}}(\hat{\boldsymbol{s}}, p(\boldsymbol{x}))$, for any $a \in (0,1)$ and a fixed function $\overline{\boldsymbol{s}}$, we have:

$$
\begin{aligned}
J_{\mathrm{DSM}}(\hat{\boldsymbol{s}}, p(\boldsymbol{x})) &= J_{\mathrm{DSM}}(\hat{\boldsymbol{s}}, p(\boldsymbol{x})) - (1+a)\hat{J}_{\mathrm{DSM}}(\hat{\boldsymbol{s}}, p(\boldsymbol{x})) + (1+a)\hat{J}_{\mathrm{DSM}}(\hat{\boldsymbol{s}}, p(\boldsymbol{x})) \\
&= J_{\mathrm{DSM}}(\hat{\boldsymbol{s}}, p(\boldsymbol{x})) - (1+a)\hat{J}_{\mathrm{DSM}}(\hat{\boldsymbol{s}}, p(\boldsymbol{x})) + (1+a) \inf_{\boldsymbol{s} \in \mathcal{S}} \hat{J}_{\mathrm{DSM}}(\boldsymbol{s}, p(\boldsymbol{x})) \\
&\leq J_{\mathrm{DSM}}(\hat{\boldsymbol{s}}, p(\boldsymbol{x})) - (1+a)\hat{J}_{\mathrm{DSM}}(\hat{\boldsymbol{s}}, p(\boldsymbol{x})) \\
&\quad + (1+a)\left(\hat{J}_{\mathrm{DSM}}(\overline{\boldsymbol{s}}, p(\boldsymbol{x})) - (1+a)J_{\mathrm{DSM}}(\overline{\boldsymbol{s}}, p(\boldsymbol{x})) + (1+a)J_{\mathrm{DSM}}(\overline{\boldsymbol{s}}, p(\boldsymbol{x}))\right) \\
&= \left(J_{\mathrm{DSM}}(\hat{\boldsymbol{s}}, p(\boldsymbol{x})) - (1+a)\hat{J}_{\mathrm{DSM}}(\hat{\boldsymbol{s}}, p(\boldsymbol{x}))\right) \\
&\quad + (1+a)\left(\hat{J}_{\mathrm{DSM}}(\overline{\boldsymbol{s}}, p(\boldsymbol{x})) - (1+a)J_{\mathrm{DSM}}(\overline{\boldsymbol{s}}, p(\boldsymbol{x}))\right) + (1+a)^2 J_{\mathrm{DSM}}(\overline{\boldsymbol{s}}, p(\boldsymbol{x})) .
\end{aligned}
$$

**First term** Firstly, we define that:

$$
j_{\mathrm{DSM}}(\boldsymbol{s}, \boldsymbol{x}, p(\boldsymbol{x})) = \mathbb{E}_{\hat{\boldsymbol{x}} \sim p(\hat{\boldsymbol{x}}|\boldsymbol{x})}\left\| \boldsymbol{s}(\hat{\boldsymbol{x}}) - \frac{\partial \log p(\hat{\boldsymbol{x}}|\boldsymbol{x})}{\partial \hat{\boldsymbol{x}}} \right\|_2^2 .
$$

For any $\boldsymbol{s} \in \mathcal{S}$, we have:

$$
\begin{aligned}
j_{\mathrm{DSM}}(\boldsymbol{s}, \boldsymbol{x}, p(\boldsymbol{x})) &= \mathbb{E}_{\hat{\boldsymbol{x}} \sim p(\hat{\boldsymbol{x}}|\boldsymbol{x})}\left\| \boldsymbol{s}(\hat{\boldsymbol{x}}) - \frac{\partial \log p(\hat{\boldsymbol{x}}|\boldsymbol{x})}{\partial \hat{\boldsymbol{x}}} \right\|_2^2 \\
&\leq 2\mathbb{E}_{\hat{\boldsymbol{x}} \sim p(\hat{\boldsymbol{x}}|\boldsymbol{x})}\left( \|\boldsymbol{s}(\hat{\boldsymbol{x}})\|_2^2 + \left\| \frac{\partial \log p(\hat{\boldsymbol{x}}|\boldsymbol{x})}{\partial \hat{\boldsymbol{x}}} \right\|_2^2 \right) \\
&= 2\mathbb{E}_{\hat{\boldsymbol{x}} \sim p(\hat{\boldsymbol{x}}|\boldsymbol{x})}\left( \|\boldsymbol{s}(\hat{\boldsymbol{x}})\|_2^2 + \left\| \frac{\boldsymbol{x} - \hat{\boldsymbol{x}}}{\sigma^2} \right\|_2^2 \right) .
\end{aligned}
\tag{12}
$$

For the first part, recall that:
$$
\hat{\boldsymbol{x}} = \boldsymbol{x} + \boldsymbol{\epsilon}, \ \boldsymbol{\epsilon} \sim \mathcal{N}(0, \sigma^2 \boldsymbol{I}) .
$$

Then we have:

$$
\left\| \frac{\hat{\boldsymbol{x}} - \boldsymbol{x}}{\sigma} \right\|_2^2 \sim \chi^2(d) .
$$

According to the Bernstein's inequality [Vershynin, 2018] and choose $t = \frac{1}{2}$, we have:

$$\mathbb{P}\left( \left| \frac{\left\| \frac{\hat{\boldsymbol{x}} - \boldsymbol{x}}{\sigma} \right\|_2^2}{d} - 1 \right| \geq \frac{1}{2} \right) \leq 2 \exp(-\frac{d}{32}) \,.$$

Then we have:

$$\mathbb{P}\left( \|\hat{\boldsymbol{x}} - \boldsymbol{x}\|_2 \geq \sigma \sqrt{\frac{3d}{2}} \right) = \mathbb{P}\left( \|\hat{\boldsymbol{x}} - \boldsymbol{x}\|_2^2 \geq \frac{3\sigma^2 d}{2} \right)$$

$$= \mathbb{P}\left( \frac{\left\| \frac{\hat{\boldsymbol{x}} - \boldsymbol{x}}{\sigma} \right\|_2^2}{d} \geq \frac{3}{2} \right)$$

$$\leq \mathbb{P}\left( \left| \frac{\left\| \frac{\hat{\boldsymbol{x}} - \boldsymbol{x}}{\sigma} \right\|_2^2}{d} - 1 \right| \geq \frac{1}{2} \right)$$

$$\leq 2 \exp(-\frac{d}{32}) \,.$$

By Lemma 5, we have:

$$\|\hat{\boldsymbol{x}}\|_2 \leq \|\hat{\boldsymbol{x}} - \boldsymbol{x}\|_2 + \|\boldsymbol{x}\|_2$$

$$\leq \sigma \sqrt{\frac{3d}{2}} + \sqrt{\sum_{i=1}^{d} (C_i + 2\sigma_i \sqrt{\log nd})^2} \,,$$

with probability at least $1 - 2\exp(-\frac{d}{32}) - \frac{1}{n^2 d}$ over the randomness of noise $\boldsymbol{\epsilon}$ and $\epsilon_i$.

Then by Lemma 5 and Nguyen et al. [2021][Lemma C.1]:

$$2\mathbb{E}_{\hat{\boldsymbol{x}} \sim p(\hat{\boldsymbol{x}}|\boldsymbol{x})} \|\boldsymbol{s}(\hat{\boldsymbol{x}})\|_2^2 \lesssim \sigma^2 d + \sum_{i=1}^{d} (C_i + 2\sigma_i \sqrt{\log nd})^2 \,. \tag{13}$$

with probability at least $1 - 2\exp(-\frac{d}{32}) - L\exp(-\Omega(m)) - \frac{1}{n^2 d}$ over the randomness of initialization $\boldsymbol{W}$, noise $\boldsymbol{\epsilon}$ and $\epsilon_i$.

For the second part:

$$2\mathbb{E}_{\hat{\boldsymbol{x}} \sim p(\hat{\boldsymbol{x}}|\boldsymbol{x})} \left\| \frac{\boldsymbol{x} - \hat{\boldsymbol{x}}}{\sigma^2} \right\|_2^2 = 2\mathbb{E}_{\boldsymbol{\epsilon} \sim \mathcal{N}(0, \sigma^2 \boldsymbol{I})} \left\| \frac{\boldsymbol{\epsilon}}{\sigma^2} \right\|_2^2$$

$$= 2\mathbb{E}_{\boldsymbol{\epsilon}' \sim \mathcal{N}(0, \boldsymbol{I})} \|\boldsymbol{\epsilon}'\|_2^2 \tag{14}$$

$$= 2\mathbb{E}_{\epsilon'' \sim \chi^2(d)} \epsilon''$$

$$= 2d \,.$$

Combine Eqs. (12) to (14), we have:

$$j_{\text{DSM}}(\boldsymbol{s}, \boldsymbol{x}, p(\boldsymbol{x})) \leq 2\mathbb{E}_{\hat{\boldsymbol{x}} \sim p(\hat{\boldsymbol{x}}|\boldsymbol{x})} \left( \|\boldsymbol{s}(\hat{\boldsymbol{x}})\|_2^2 + \left\| \frac{\boldsymbol{x} - \hat{\boldsymbol{x}}}{\sigma^2} \right\|_2^2 \right)$$

$$\lesssim (\sigma^2 + 2)d + \sum_{i=1}^{d} (C_i + 2\sigma_i \sqrt{\log nd})^2 \,, \tag{15}$$

with probability at least $1 - 2\exp(-\frac{d}{32}) - L\exp(-\Omega(m)) - \frac{1}{n^2 d}$ over the randomness of initialization $\boldsymbol{W}$, noise $\boldsymbol{\epsilon}$ and $\epsilon_i$.

According to the Bernstein-type concentration inequality Chen et al. [2023a][Lemma 15], for $\delta \in (0,1)$, $a \leq 1$ and $\tau > 0$, we have:

$$J_{\text{DSM}}(\hat{\boldsymbol{s}}, p(\boldsymbol{x})) - (1+a)\hat{J}_{\text{DSM}}(\hat{\boldsymbol{s}}, p(\boldsymbol{x})) \lesssim \frac{1+3/a}{2n}\big((\sigma^2+2)d + \sum_{i=1}^{d}(C_i + 2\sigma_i\sqrt{\log nd})^2\big)\log\frac{\mathcal{N}_c(\tau, \mathcal{S})}{\delta} + (2+a)\tau,$$

with probability at least $1 - \delta - 2\exp(-\frac{d}{32}) - L\exp(-\Omega(m)) - \frac{1}{nd}$ over the randomness of initialization $\boldsymbol{W}$, noise $\boldsymbol{\epsilon}$ and $\epsilon_i$.

**Second term** According to the Bernstein-type concentration inequality Chen et al. [2023a][Lemma 15] and Eq. (15), for $\delta \in (0,1)$, $a \leq 1$, $\tau > 0$ and a fixed function $\overline{\boldsymbol{s}}$, , we have:

$$\hat{J}_{\text{DSM}}(\overline{\boldsymbol{s}}, p(\boldsymbol{x})) - (1+a)J_{\text{DSM}}(\overline{\boldsymbol{s}}, p(\boldsymbol{x})) \lesssim \frac{1+3/a}{2n}\big((\sigma^2+2)d + \sum_{i=1}^{d}(C_i + 2\sigma_i\sqrt{\log nd})^2\big)\log\frac{1}{\delta} + (2+a)\tau,$$

with probability at least $1 - \delta - 2\exp(-\frac{d}{32}) - L\exp(-\Omega(m)) - \frac{1}{nd}$ over the randomness of initialization $\boldsymbol{W}$, noise $\boldsymbol{\epsilon}$ and $\epsilon_i$.

**Third term** We can derive that:

$$J_{\text{DSM}}(\overline{\boldsymbol{s}}, p(\boldsymbol{x})) = J_{\text{ESM}}(\overline{\boldsymbol{s}}, p(\boldsymbol{x})) + J_{\text{DSM}}(\overline{\boldsymbol{s}}, p(\boldsymbol{x})) - J_{\text{ESM}}(\overline{\boldsymbol{s}}, p(\boldsymbol{x})).$$

According to Lemma 3, since the error term is invariant with respect to translations on $\nabla \log p(\cdot)$ and the homogeneity of the ReLU neural network, we can omit $\nabla \log p(\boldsymbol{0}) = 0$ and rescale bound for the input data, for any $\varepsilon \in (0,1)$, there exists an approximation function $\overline{\boldsymbol{s}}$ satisfying $\|\nabla \log p(\cdot) - \overline{\boldsymbol{s}}(\cdot)\|_\infty \leq \varepsilon$, then we have:

$$J_{\text{ESM}}(\overline{\boldsymbol{s}}, p(\boldsymbol{x})) \leq \frac{d\varepsilon^2}{2},$$

with probability at least $1 - \frac{1}{nd}$ over the randomness of noise $\epsilon_i$ and satisfy the configuration of network architecture in Lemma 3.

According to Vincent [2011], we have:

$$J_{\text{DSM}}(\overline{\boldsymbol{s}}, p(\boldsymbol{x})) - J_{\text{ESM}}(\overline{\boldsymbol{s}}, p(\boldsymbol{x})) = \frac{1}{2}\mathbb{E}_{\boldsymbol{x}}\mathbb{E}_{\hat{\boldsymbol{x}}\sim\phi(\boldsymbol{x}|\boldsymbol{x})}\big[\|\nabla_{\hat{\boldsymbol{x}}}\log\phi(\boldsymbol{x}|\boldsymbol{x})\|_2^2\big] - \frac{1}{2}\|\nabla\log p(\cdot)\|_{\ell^2(p)}^2.$$

which is an absolute value that does not depend on $\boldsymbol{s}$. So we can define that:

$$E_1 := \frac{1}{2}\mathbb{E}_{\boldsymbol{x}}\mathbb{E}_{\hat{\boldsymbol{x}}\sim\phi(\boldsymbol{x}|\boldsymbol{x})}\big[\|\nabla_{\hat{\boldsymbol{x}}}\log\phi(\boldsymbol{x}|\boldsymbol{x})\|_2^2\big] - \frac{1}{2}\|\nabla\log p(\cdot)\|_{\ell^2(p)}^2.$$

So if we choose $\overline{\boldsymbol{s}}$ is the approximation function that Lemma 3 provide, then we have:

$$J_{\text{DSM}}(\overline{\boldsymbol{s}}, p(\boldsymbol{x})) \leq \frac{d\varepsilon^2}{2} + E_1.$$

**Putting things together**   Combine all three terms, we have:

$$
\begin{aligned}
J_{\text{DSM}}(\hat{\boldsymbol{s}}, p(\boldsymbol{x})) \leq & \left( J_{\text{DSM}}(\hat{\boldsymbol{s}}, p(\boldsymbol{x})) - (1+a)\hat{J}_{\text{DSM}}(\hat{\boldsymbol{s}}, p(\boldsymbol{x})) \right) \\
& + (1+a)\left( \hat{J}_{\text{DSM}}(\overline{\boldsymbol{s}}, p(\boldsymbol{x})) - (1+a)J_{\text{DSM}}(\overline{\boldsymbol{s}}, p(\boldsymbol{x})) \right) + (1+a)^2 J_{\text{DSM}}(\overline{\boldsymbol{s}}, p(\boldsymbol{x})) \\
\lesssim & \left( J_{\text{DSM}}(\hat{\boldsymbol{s}}, p(\boldsymbol{x})) - (1+a)\hat{J}_{\text{DSM}}(\hat{\boldsymbol{s}}, p(\boldsymbol{x})) \right) \\
& + (1+a)\left( \hat{J}_{\text{DSM}}(\overline{\boldsymbol{s}}, p(\boldsymbol{x})) - (1+a)J_{\text{DSM}}(\overline{\boldsymbol{s}}, p(\boldsymbol{x})) \right) + (1+a)^2 \left( \frac{d\varepsilon^2}{2} + E_1 \right) \\
= & \left( J_{\text{DSM}}(\hat{\boldsymbol{s}}, p(\boldsymbol{x})) - (1+a)\hat{J}_{\text{DSM}}(\hat{\boldsymbol{s}}, p(\boldsymbol{x})) \right) \\
& + (1+a)\left( \hat{J}_{\text{DSM}}(\overline{\boldsymbol{s}}, p(\boldsymbol{x})) - (1+a)J_{\text{DSM}}(\overline{\boldsymbol{s}}, p(\boldsymbol{x})) \right) + (1+a)^2 \frac{d\varepsilon^2}{2} + (2a+a^2)E_1 + E_1 \,.
\end{aligned}
$$

Then:

$$
\begin{aligned}
J_{\text{ESM}}(\hat{\boldsymbol{s}}, p(\boldsymbol{x})) = & \, J_{\text{DSM}}(\hat{\boldsymbol{s}}, p(\boldsymbol{x})) - E_1 \\
\lesssim & \left( J_{\text{DSM}}(\hat{\boldsymbol{s}}, p(\boldsymbol{x})) - (1+a)\hat{J}_{\text{DSM}}(\hat{\boldsymbol{s}}, p(\boldsymbol{x})) \right) \\
& + (1+a)\left( \hat{J}_{\text{DSM}}(\overline{\boldsymbol{s}}, p(\boldsymbol{x})) - (1+a)J_{\text{DSM}}(\overline{\boldsymbol{s}}, p(\boldsymbol{x})) \right) + (1+a)^2 \frac{d\varepsilon^2}{2} + (2a+a^2)E_1 \\
\lesssim & \, \frac{1+3/a}{2n}\left( (\sigma^2+2)d + \sum_{i=1}^{d}(C_i + 2\sigma_i\sqrt{\log nd})^2 \right) \log \frac{\mathcal{N}_c(\tau, \mathcal{S})}{\delta} + (2+a)\tau \\
& + (1+a)\left( \frac{1+3/a}{2n}\left( (\sigma^2+2)d + \sum_{i=1}^{d}(C_i + 2\sigma_i\sqrt{\log nd})^2 \right) \log \frac{1}{\delta} + (2+a)\tau \right) \\
& + (1+a)^2 \frac{d\varepsilon^2}{2} + (2a+a^2)E_1 \,,
\end{aligned}
$$

with probability at least $1 - 2\delta - 4\exp(-\frac{d}{32}) - 2L\exp(-\Omega(m)) - \frac{1}{nd}$ over the randomness of initialization $\boldsymbol{W}$, noise $\boldsymbol{\epsilon}$ and $\epsilon_i$.

Let $a = \varepsilon^2$, $\tau = \frac{1}{n}$, $\sigma_i \approx \sigma$ and $\frac{C_i}{\sigma_i} \approx 1$, $\forall i \in [d]$. Then we have:

$$
J_{\text{ESM}}(\hat{\boldsymbol{s}}, p(\boldsymbol{x})) \lesssim \frac{\sigma^2 d \log nd}{n\varepsilon^2} \log \frac{\mathcal{N}_c(\frac{1}{n}, \mathcal{S})}{\delta} + \frac{1}{n} + d\varepsilon^2 \,,
$$

with probability at least $1 - 2\delta - 4\exp(-\frac{d}{32}) - 2L\exp(-\Omega(m)) - \frac{1}{nd}$ over the randomness of initialization $\boldsymbol{W}$, noise $\boldsymbol{\epsilon}$ and $\epsilon_i$. $\qquad\square$

# F   Proof of the error bound of topological ordering using the SCORE algorithm in a causal model (Theorem 2)

*Proof.* We set the weights of the neural network after training are $\widehat{\boldsymbol{W}}$. i.e.

$$
\boldsymbol{s}(\boldsymbol{x}) = \widehat{\boldsymbol{W}}_L \phi(\widehat{\boldsymbol{W}}_{L-1} \cdots \phi(\widehat{\boldsymbol{W}}_1 \boldsymbol{x}) \cdots) \,.
$$

According to the standard chain rule and Zhu et al. [2022][Lemma 3], we have:

$$
\nabla_{\boldsymbol{x}} \boldsymbol{s}(\boldsymbol{x})^\top = \widehat{\boldsymbol{W}}_L \widehat{\boldsymbol{D}}_{L-1} \widehat{\boldsymbol{W}}_{L-1} \cdots \widehat{\boldsymbol{D}}_1 \widehat{\boldsymbol{W}}_1 \,.
$$

Let $\boldsymbol{v}_i$ be a one-hot vector with length $d$, with the $i$-th element is 1 and the rest of the elements are 0, then we have:

$$
\begin{aligned}
\frac{\partial s_i(\boldsymbol{x})}{\partial x^{(i)}} &= \boldsymbol{v}_i \widehat{\boldsymbol{W}}_L \widehat{\boldsymbol{D}}_{L-1} \widehat{\boldsymbol{W}}_{L-1} \cdots \widehat{\boldsymbol{D}}_1 \widehat{\boldsymbol{W}}_1 \boldsymbol{v}_i \\
&\leq \|\boldsymbol{v}_i\|_2 \left\| \widehat{\boldsymbol{W}}_L \widehat{\boldsymbol{D}}_{L-1} \widehat{\boldsymbol{W}}_{L-1} \cdots \widehat{\boldsymbol{D}}_1 \right\|_2 \left\| \widehat{\boldsymbol{W}}_1 \right\|_2 \|\boldsymbol{v}_i\|_2 \\
&= \left\| \widehat{\boldsymbol{W}}_L \widehat{\boldsymbol{D}}_{L-1} \widehat{\boldsymbol{W}}_{L-1} \cdots \widehat{\boldsymbol{D}}_1 \right\|_2 \left\| \widehat{\boldsymbol{W}}_1 \right\|_2 \\
&= \left( \|\boldsymbol{W}_L \boldsymbol{D}_{L-1} \boldsymbol{W}_{L-1} \cdots \boldsymbol{D}_1\|_2 + \left\| \widehat{\boldsymbol{W}}_L \widehat{\boldsymbol{D}}_{L-1} \widehat{\boldsymbol{W}}_{L-1} \cdots \widehat{\boldsymbol{D}}_1 - \boldsymbol{W}_L \boldsymbol{D}_{L-1} \boldsymbol{W}_{L-1} \cdots \boldsymbol{D}_1 \right\|_2 \right) \\
&\quad \times \left( \|\boldsymbol{W}_1\|_2 + \left\| \widehat{\boldsymbol{W}}_1 - \boldsymbol{W}_1 \right\|_2 \right) \\
&\coloneqq (T_1 + T_2) \times (T_3 + T_4) .
\end{aligned}
\tag{16}
$$

Firstly, we focus on $T_1$. Define $\boldsymbol{t}_l(\boldsymbol{v}) = \boldsymbol{D}_l \boldsymbol{W}_l \cdots \boldsymbol{D}_1 \boldsymbol{v}$, then for any vector $\boldsymbol{v}$ that satisfy $\|\boldsymbol{v}\|_2 = 1$:

$$
\begin{aligned}
\|\boldsymbol{W}_L \boldsymbol{D}_{L-1} \boldsymbol{W}_{L-1} \cdots \boldsymbol{D}_1 \boldsymbol{v}\|_2 &= \|\boldsymbol{W}_L \boldsymbol{t}_{L-1}(\boldsymbol{v})\|_2 \\
&= \sqrt{\|\boldsymbol{W}_L \boldsymbol{t}_{L-1}(\boldsymbol{v})\|_2^2} \\
&= \sqrt{\frac{\|\boldsymbol{W}_L \boldsymbol{t}_{L-1}(\boldsymbol{v})\|_2^2}{\|\boldsymbol{t}_{L-1}(\boldsymbol{v})\|_2^2} \frac{\|\boldsymbol{t}_{L-1}(\boldsymbol{v})\|_2^2}{\|\boldsymbol{t}_{L-2}(\boldsymbol{v})\|_2^2} \cdots \frac{\|\boldsymbol{t}_2(\boldsymbol{v})\|_2^2}{\|\boldsymbol{t}_1(\boldsymbol{v})\|_2^2} \|\boldsymbol{t}_1(\boldsymbol{v})\|_2^2} .
\end{aligned}
\tag{17}
$$

According to Zhu et al. [2022][Lemma 2], we have:

$$
\frac{\|\boldsymbol{t}_l(\boldsymbol{v})\|_2^2}{\|\boldsymbol{t}_{l-1}(\boldsymbol{v})\|_2^2} \sim \frac{2}{m} \chi^2(\varrho), \quad \forall l = 2, \cdots, L-1 ,
$$

where $\varrho \sim \mathrm{Ber}(m, 1/2)$.

According to Ghosh [2021], with probability at least $1 - \exp(-\Theta(m))$ over the randomness of initialization $\boldsymbol{W}_l$, we have:

$$
\frac{\|\boldsymbol{t}_l(\boldsymbol{v})\|_2^2}{\|\boldsymbol{t}_{l-1}(\boldsymbol{v})\|_2^2} \leq 4, \quad \forall l = 2, \cdots, L-1 .
\tag{18}
$$

By the definition of chi-square distribution, we have:

$$
\frac{\|\boldsymbol{W}_L \boldsymbol{t}_{L-1}\|_2^2}{\|\boldsymbol{t}_{L-1}\|_2^2} \sim \frac{\chi^2(d)}{d} ,
$$

Similar, according to Ghosh [2021], with probability at least $1 - \exp(-\Theta(d))$ over the randomness of initialization $\boldsymbol{W}_L$, we have:

$$
\frac{\|\boldsymbol{W}_L \boldsymbol{t}_{L-1}\|_2^2}{\|\boldsymbol{t}_{L-1}\|_2^2} \leq 2 .
\tag{19}
$$

And we can derive that:

$$
\|\boldsymbol{t}_1(\boldsymbol{v})\|_2^2 = \|\boldsymbol{D}_1 \boldsymbol{v}\|_2^2 \leq \left( \|\boldsymbol{D}_1\|_2 \|\boldsymbol{v}\|_2 \right)^2 \leq 1 .
\tag{20}
$$

Combine Eqs. (17) to (20), we have:

$$\|\boldsymbol{W}_L \boldsymbol{D}_{L-1} \boldsymbol{W}_{L-1} \cdots \boldsymbol{D}_1 \boldsymbol{v}\|_2 = \sqrt{\frac{\|\boldsymbol{W}_L \boldsymbol{t}_{L-1}(\boldsymbol{v})\|_2^2}{\|\boldsymbol{t}_{L-1}(\boldsymbol{v})\|_2^2} \frac{\|\boldsymbol{t}_{L-1}(\boldsymbol{v})\|_2^2}{\|\boldsymbol{t}_{L-2}(\boldsymbol{v})\|_2^2} \cdots \frac{\|\boldsymbol{t}_2(\boldsymbol{v})\|_2^2}{\|\boldsymbol{t}_1(\boldsymbol{v})\|_2^2} \|\boldsymbol{t}_1(\boldsymbol{v})\|_2^2} \leq 2^{\frac{2L-1}{2}} ,$$

with probability at least $1 - \exp(-\Theta(d)) - (L-2)\exp(-\Theta(m))$ over the randomness of initialization $\boldsymbol{W}$.

i.e.

$$T_1 = \|\boldsymbol{W}_L \boldsymbol{D}_{L-1} \boldsymbol{W}_{L-1} \cdots \boldsymbol{D}_1\|_2 \leq 2^{\frac{2L-1}{2}} , \tag{21}$$

with probability at least $1 - \exp(-\Theta(d)) - (L-2)\exp(-\Theta(m))$ over the randomness of initialization $\boldsymbol{W}$.

For a perturbation matrices satisfy $T_4 = \left\|\widehat{\boldsymbol{W}}_l - \boldsymbol{W}_l\right\|_2 \leq \omega = \mathcal{O}(\frac{1}{L^{3/2}})$, $\forall l \in [L]$, by Allen-Zhu et al. [2019, Lemma 8.7], we obtain that for any integer $s = \mathcal{O}(m\omega^{2/3}L)$ and $d \leq \mathcal{O}(\frac{m}{L\log m})$, with probability at least $1 - \exp\left(-\Omega(m\log m\omega^{2/3}L)\right)$ over the randomness of initialization $\boldsymbol{W}$, it holds that:

$$T_2 = \left\|\widehat{\boldsymbol{W}}_L \widehat{\boldsymbol{D}}_{L-1} \widehat{\boldsymbol{W}}_{L-1} \cdots \widehat{\boldsymbol{D}}_1 - \boldsymbol{W}_L \boldsymbol{D}_{L-1} \boldsymbol{W}_{L-1} \cdots \boldsymbol{D}_1\right\|_2 \leq \mathcal{O}\left(\frac{\omega^{1/3}L^2\sqrt{m\log m}}{\sqrt{d}}\right). \tag{22}$$

For $T_3$, according to Lemma 4, we have that for every $t \geq 0$, with probability at least $1 - 2\exp(-t^2/2)$ over the randomness of initialization $\boldsymbol{W}_1$, one has:

$$T_3 = \|\boldsymbol{W}_1\|_2 \leq \sqrt{\frac{2}{m}}(\sqrt{m} + \sqrt{d} + t). \tag{23}$$

Combine Eqs. (16) and (21) to (23), choose $t = \sqrt{m}$ we have:

$$\begin{aligned}
\frac{\partial s_i(\boldsymbol{x})}{\partial x^{(i)}} &\leq (T_1 + T_2) \times (T_3 + T_4) \\
&\lesssim \left(2^{\frac{2L-1}{2}} + \frac{\omega^{1/3}L^2\sqrt{m\log m}}{\sqrt{d}}\right) \times \left(\frac{1}{L^{3/2}} + \sqrt{\frac{2}{m}}(2\sqrt{m} + \sqrt{d})\right) \\
&\lesssim \frac{2^L\sqrt{\log m}(\sqrt{m} + \sqrt{d})}{\sqrt{d}} ,
\end{aligned} \tag{24}$$

with probability at least $1 - \exp(-\Theta(d)) - L\exp(-\Theta(m)) - \exp\left(-\Omega(m\log m)\right)$ over the randomness of initialization $\boldsymbol{W}$.

Then, for $\left(\frac{\partial s_i(\boldsymbol{x})}{\partial x^{(i)}}\right)$, we have that:

$$\left(\frac{\partial s_i(\boldsymbol{x})}{\partial x^{(i)}}\right)^2 \lesssim \frac{2^{2L}\log m(m+d)}{d} , \tag{25}$$

with probability at least $1 - \exp(-\Theta(d)) - L\exp(-\Theta(m)) - \exp\left(-\Omega(m\log m)\right)$ over the randomness of initialization $\boldsymbol{W}$.

According to Hoeffding's inequality for bounded random variables [Vershynin, 2018][Thmorem 2.2.6], we have that:

$$\left|\frac{1}{n}\sum_{i=1}^n \frac{\partial s_i(\boldsymbol{x})}{\partial x^{(i)}} - \mathbb{E}\frac{\partial s_i(\boldsymbol{x})}{\partial x^{(i)}}\right| \leq \frac{C_m}{12\mathbb{E}\frac{\partial s_i(\boldsymbol{x})}{\partial x^{(i)}}} ,$$

with probability at least $1 - \exp(-\Theta(d)) - L\exp(-\Theta(m)) - \exp\left(-\Omega(m\log m)\right) - 2\exp(-\Omega(\frac{nC_m^2 d^2}{2^{4L+5}(\log m)^2(m^2+d^2)}))$, and

$$\left| \frac{1}{n}\sum_{i=1}^n \left( \frac{\partial s_i(\boldsymbol{x})}{\partial x^{(i)}} \right)^2 - \mathbb{E}\left( \frac{\partial s_i(\boldsymbol{x})}{\partial x^{(i)}} \right)^2 \right| \leq \frac{C_m}{4},$$

with probability at least $1 - \exp(-\Theta(d)) - L\exp(-\Theta(m)) - \exp\left(-\Omega(m\log m)\right) - 2\exp(-\frac{nC_m^2 d^2}{2^{4L+5}(\log m)^2(m^2+d^2)})$.

Then we have:

$$
\begin{aligned}
\left| \mathrm{Var}\left( \frac{\partial s_i(\boldsymbol{x})}{\partial x^{(i)}} \right) - \hat{\mathrm{Var}}\left( \frac{\partial s_i(\boldsymbol{x})}{\partial x^{(i)}} \right) \right| &= \left| \mathbb{E}\left( \frac{\partial s_i(\boldsymbol{x})}{\partial x^{(i)}} \right)^2 - \left( \mathbb{E}\frac{\partial s_i(\boldsymbol{x})}{\partial x^{(i)}} \right)^2 - \sum_{i=1}^n \left( \frac{\partial s_i(\boldsymbol{x})}{\partial x^{(i)}} \right)^2 + \left( \frac{1}{n}\sum_{i=1}^n \frac{\partial s_i(\boldsymbol{x})}{\partial x^{(i)}} \right)^2 \right| \\
&\leq \left| \mathbb{E}\left( \frac{\partial s_i(\boldsymbol{x})}{\partial x^{(i)}} \right)^2 - \sum_{i=1}^n \left( \frac{\partial s_i(\boldsymbol{x})}{\partial x^{(i)}} \right)^2 \right| + \left| -\left( \mathbb{E}\frac{\partial s_i(\boldsymbol{x})}{\partial x^{(i)}} \right)^2 + \left( \frac{1}{n}\sum_{i=1}^n \frac{\partial s_i(\boldsymbol{x})}{\partial x^{(i)}} \right)^2 \right| \\
&\leq \frac{C_m}{4} + \left| \frac{1}{n}\sum_{i=1}^n \frac{\partial s_i(\boldsymbol{x})}{\partial x^{(i)}} - \mathbb{E}\frac{\partial s_i(\boldsymbol{x})}{\partial x^{(i)}} \right| \left| \mathbb{E}\frac{\partial s_i(\boldsymbol{x})}{\partial x^{(i)}} + \frac{1}{n}\sum_{i=1}^n \frac{\partial s_i(\boldsymbol{x})}{\partial x^{(i)}} \right| \\
&\leq \frac{C_m}{2},
\end{aligned}
$$

$\square$

with probability at least $1 - \exp(-\Theta(d)) - L\exp(-\Theta(m)) - \exp\left(-\Theta(m\log m)\right) - 2\exp(-\frac{nC_m^2 d^2}{2^{4L+5}(\log m)^2(m^2+d^2)})$.

Thus, for $i$ is a leaf and $j$ is not a leaf, according to Assumption 2 and Lemma 1, we have:

$$\mathrm{Var}\left( \frac{\partial s_j(\boldsymbol{x})}{\partial x^{(j)}} \right) - \mathrm{Var}\left( \frac{\partial s_i(\boldsymbol{x})}{\partial x^{(i)}} \right) \geq C_m.$$

Then:

$$
\begin{aligned}
\hat{\mathrm{Var}}\left( \frac{\partial s_i(\boldsymbol{x})}{\partial x^{(i)}} \right) &= \hat{\mathrm{Var}}\left( \frac{\partial s_i(\boldsymbol{x})}{\partial x^{(i)}} \right) - \mathrm{Var}\left( \frac{\partial s_i(\boldsymbol{x})}{\partial x^{(i)}} \right) + \mathrm{Var}\left( \frac{\partial s_i(\boldsymbol{x})}{\partial x^{(i)}} \right) \\
&\leq \frac{C_m}{2} + \mathrm{Var}\left( \frac{\partial s_i(\boldsymbol{x})}{\partial x^{(i)}} \right) \\
&\leq \mathrm{Var}\left( \frac{\partial s_j(\boldsymbol{x})}{\partial x^{(j)}} \right) - \frac{C_m}{2} \\
&= \mathrm{Var}\left( \frac{\partial s_j(\boldsymbol{x})}{\partial x^{(j)}} \right) - \hat{\mathrm{Var}}\left( \frac{\partial s_j(\boldsymbol{x})}{\partial x^{(j)}} \right) + \hat{\mathrm{Var}}\left( \frac{\partial s_j(\boldsymbol{x})}{\partial x^{(j)}} \right) - \frac{C_m}{2} \\
&\leq \frac{C_m}{2} + \hat{\mathrm{Var}}\left( \frac{\partial s_j(\boldsymbol{x})}{\partial x^{(j)}} \right) - \frac{C_m}{2} \\
&= \hat{\mathrm{Var}}\left( \frac{\partial s_j(\boldsymbol{x})}{\partial x^{(j)}} \right).
\end{aligned}
$$

with probability at least $1 - \exp(-\Theta(d)) - L\exp(-\Theta(m)) - \exp\left(-\Theta(m\log m)\right) - 2\exp(-\frac{nC_m^2 d^2}{2^{4L+5}(\log m)^2(m^2+d^2)})$. Considering all variables, then with probability at least:

$$1 - \exp(-\Theta(d)) - (L+1)\exp(-\Theta(m)) - 2n\exp(-\frac{nC_m^2 d^2}{2^{4L+5}(\log m)^2(m^2+d^2)}),$$

that Algorithm 1 can completely recover the correct topological order of the non-linear additive Gaussian noise model.

# G Proof of the error bound of score function estimate for the score-based generative modeling (Theorem 3)

*Proof.* Firstly, we use oracle inequality to decompose $\mathcal{L}(\hat{s})$, for any $a \in (0, 1)$ and a fixed function $\bar{s}$, we have:

$$
\begin{aligned}
\mathcal{L}(\hat{s}) &= \mathcal{L}(\hat{s}) - (1+a)\hat{\mathcal{L}}(\hat{s}) + (1+a)\hat{\mathcal{L}}(\hat{s}) \\
&= \mathcal{L}(\hat{s}) - (1+a)\hat{\mathcal{L}}(\hat{s}) + (1+a)\inf_{s \in \mathcal{S}} \hat{\mathcal{L}}(s) \\
&\leq \mathcal{L}(\hat{s}) - (1+a)\hat{\mathcal{L}}(\hat{s}) + (1+a)\big(\hat{\mathcal{L}}(\bar{s}) - (1+a)\mathcal{L}(\bar{s}) + (1+a)\mathcal{L}(\bar{s})\big) \\
&= \Big(\mathcal{L}(\hat{s}) - (1+a)\hat{\mathcal{L}}(\hat{s})\Big) + (1+a)\Big(\hat{\mathcal{L}}(\bar{s}) - (1+a)\mathcal{L}(\bar{s})\Big) + (1+a)^2\mathcal{L}(\bar{s}).
\end{aligned}
$$

**First term** For any $s \in \mathcal{S}$, we have:

$$
\begin{aligned}
\ell(\boldsymbol{x}; \boldsymbol{s}) &= \frac{1}{T - t_0} \int_{t_0}^{T} \mathbb{E}_{\boldsymbol{x}_t \sim p_{0t}(\boldsymbol{x}_t | \boldsymbol{x}_0 = \boldsymbol{x})} \big[ \| \nabla_{\boldsymbol{x}_t} \log p_{0t}(\boldsymbol{x}_t | \boldsymbol{x}_0 = \boldsymbol{x}) - \boldsymbol{s}(\boldsymbol{x}_t, t) \|_2^2 \big] \mathrm{d}t \\
&= \frac{1}{T - t_0} \int_{t_0}^{T} \mathbb{E}_{\boldsymbol{x}_t \sim p_{0t}(\boldsymbol{x}_t | \boldsymbol{x}_0 = \boldsymbol{x})} \left( \left\| \frac{\boldsymbol{x}_t - \alpha(t)\boldsymbol{x}}{h(t)} + \boldsymbol{s}(\boldsymbol{x}_t, t) \right\|_2^2 \right) \mathrm{d}t \\
&\leq \frac{3}{T - t_0} \int_{t_0}^{T} \mathbb{E}_{\boldsymbol{x}_t \sim p_{0t}(\boldsymbol{x}_t | \boldsymbol{x}_0 = \boldsymbol{x})} \left[ \left( \left\| \frac{\boldsymbol{x}_t}{h(t)} \right\|_2^2 + \left\| \frac{\alpha(t)\boldsymbol{x}}{h(t)} \right\|_2^2 + \| \boldsymbol{s}(\boldsymbol{x}_t, t) \|_2^2 \right) \right] \mathrm{d}t \\
&= \frac{3}{T - t_0} \int_{t_0}^{T} \mathbb{E}_{\boldsymbol{x}_t \sim p_{0t}(\boldsymbol{x}_t | \boldsymbol{x}_0 = \boldsymbol{x})} \left( \left\| \frac{\boldsymbol{x}_t}{h(t)} \right\|_2^2 \right) \mathrm{d}t \\
&\quad + \frac{3}{T - t_0} \int_{t_0}^{T} \left( \left\| \frac{\alpha(t)\boldsymbol{x}}{h(t)} \right\|_2^2 \right) \mathrm{d}t \\
&\quad + \frac{3}{T - t_0} \int_{t_0}^{T} \mathbb{E}_{\boldsymbol{x}_t \sim p_{0t}(\boldsymbol{x}_t | \boldsymbol{x}_0 = \boldsymbol{x})} \left( \| \boldsymbol{s}(\boldsymbol{x}_t, t) \|_2^2 \right) \mathrm{d}t .
\end{aligned}
\tag{26}
$$

For the first part, for forward process SDE Eq. (6) we can easily derive that $p_{0t}(\boldsymbol{x}_t | \boldsymbol{x}_0) \sim \mathcal{N}\big(\alpha(t)\boldsymbol{x}_0, h(t)I_d\big)$, where $\alpha(t) = e^{-\frac{t}{2}}$ and $h(t) = 1 - \alpha(t)^2$.

$$
\begin{aligned}
&\int_{t_0}^{T} \mathbb{E}_{\boldsymbol{x}_t \sim p_{0t}(\boldsymbol{x}_t | \boldsymbol{x}_0 = \boldsymbol{x})} \left( \left\| \frac{\boldsymbol{x}_t}{h(t)} \right\|_2^2 \right) \mathrm{d}t \\
&= \int_{t_0}^{T} \mathbb{E}_{\boldsymbol{x}_t \sim \mathcal{N}\big(\alpha(t)\boldsymbol{x}_0, h(t)I_d\big)} \left( \left\| \frac{\boldsymbol{x}_t}{h(t)} \right\|_2^2 \right) \mathrm{d}t \\
&= \int_{t_0}^{T} \left( \sum_{i=1}^{d} \big[ \mathbb{E}_{x_t^{(i)} \sim \mathcal{N}\big(\alpha(t)x_0^{(i)}, h(t)\big)} \Big( \frac{x_t^{(i)}}{h(t)} \Big)^2 \big] \right) \mathrm{d}t \\
&= \int_{t_0}^{T} \left( \sum_{i=1}^{d} \big[ \frac{\alpha(t)^2}{h(t)^2}(x_0^{(i)})^2 + \frac{1}{h(t)} \big] \right) \mathrm{d}t \\
&= \sum_{i=1}^{d} (x_0^{(i)})^2 \int_{t_0}^{T} \frac{\alpha(t)^2}{h(t)^2} \mathrm{d}t + \int_{t_0}^{T} \frac{d}{h(t)} \mathrm{d}t \\
&\leq \frac{T - t_0}{T t_0} C_d^2 + d(T - \log(t_0)) .
\end{aligned}
\tag{27}
$$

For the second part:

$$\int_{t_0}^{T} \left\| \frac{\alpha(t)\boldsymbol{x}}{h(t)} \right\|_2^2 \mathrm{d}t \le C_d^2 \int_{t_0}^{T} \frac{\alpha(t)^2}{h(t)^2} \mathrm{d}t = C_d^2 \frac{T - t_0}{T t_0} \,. \tag{28}$$

For the third part, by Nguyen et al. [2021][Lemma C.1]:

$$\int_{t_0}^{T} \mathbb{E}_{\boldsymbol{x}_t \sim p_{0t}(\boldsymbol{x}_t | \boldsymbol{x}_0 = \boldsymbol{x})} \left( \|\boldsymbol{s}(\boldsymbol{x}_t, t)\|_2^2 \right) \mathrm{d}t \asymp C_d^2 (T - t_0) \,, \tag{29}$$

with probability at least $1 - L \exp(-\Omega(m))$ over the randomness of initialization $\boldsymbol{W}$.

Combine Eqs. (26) to (29), we have:

$$\begin{aligned}
\ell(\boldsymbol{x}; \boldsymbol{s}) &= \frac{3}{T - t_0} \int_{t_0}^{T} \mathbb{E}_{\boldsymbol{x}_t \sim p_{0t}(\boldsymbol{x}_t | \boldsymbol{x}_0 = \boldsymbol{x})} \left( \left\| \frac{\boldsymbol{x}_t}{h(t)} \right\|_2^2 \right) \mathrm{d}t \\
&\quad + \frac{3}{T - t_0} \int_{t_0}^{T} \left( \left\| \frac{\alpha(t)\boldsymbol{x}}{h(t)} \right\|_2^2 \right) \mathrm{d}t \\
&\quad + \frac{3}{T - t_0} \int_{t_0}^{T} \mathbb{E}_{\boldsymbol{x}_t \sim p_{0t}(\boldsymbol{x}_t | \boldsymbol{x}_0 = \boldsymbol{x})} \left( \|\boldsymbol{s}(\boldsymbol{x}_t, t)\|_2^2 \right) \mathrm{d}t \\
&\lesssim \frac{3}{T - t_0} \left( \frac{T - t_0}{T t_0} C_d^2 + d(T - \log(t_0)) + C_d^2 \frac{T - t_0}{T t_0} + C_d^2 (T - t_0) \right) \\
&= \frac{3d(T - \log(t_0))}{T - t_0} + 3C_d^2 + \frac{6C_d^2}{T t_0} \,,
\end{aligned} \tag{30}$$

with probability at least $1 - L \exp(-\Omega(m))$ over the randomness of initialization $\boldsymbol{W}$.

According to the Bernstein-type concentration inequality Chen et al. [2023a][Lemma 15], for $\delta \in (0, 1)$, $a \le 1$ and $\tau > 0$, we have:

$$\mathcal{L}(\hat{\boldsymbol{s}}) - (1 + a)\hat{\mathcal{L}}(\hat{\boldsymbol{s}}) \lesssim \frac{1 + 3/a}{n} \left( \frac{d(T - \log(t_0))}{T - t_0} + C_d^2 \right) \log \frac{\mathcal{N}_c(\tau, \mathcal{S})}{\delta} + (2 + a)\tau \,,$$

with probability at least $1 - \delta - L \exp(-\Omega(m))$ over the randomness of initialization $\boldsymbol{W}$.

**Second term** According to the Bernstein-type concentration inequality Chen et al. [2023a][Lemma 15] and Eq. (30), for $\delta \in (0, 1)$, $\tau > 0$ and a fixed function $\overline{\boldsymbol{s}}$, , we have:

$$\hat{\mathcal{L}}(\overline{\boldsymbol{s}}) - (1 + a)\mathcal{L}(\overline{\boldsymbol{s}}) \lesssim \frac{1 + 3/a}{n} \left( \frac{d(T - \log(t_0))}{T - t_0} + C_d^2 \right) \log \frac{1}{\delta} + (2 + a)\tau \,,$$

with probability at least $1 - \delta - L \exp(-\Omega(m))$ over the randomness of initialization $\boldsymbol{W}$.

**Third term** We can derive that:

$$\begin{aligned}
\mathcal{L}(\overline{\boldsymbol{s}}) &= \frac{1}{T - t_0} \int_{t_0}^{T} \|\nabla \log p_t(\cdot) - \overline{\boldsymbol{s}}(\cdot, t)\|_{\ell^2(p_t)}^2 \mathrm{d}t \\
&\quad + \mathcal{L}(\overline{\boldsymbol{s}}) - \frac{1}{T - t_0} \int_{t_0}^{T} \|\nabla \log p_t(\cdot) - \overline{\boldsymbol{s}}(\cdot, t)\|_{\ell^2(p_t)}^2 \mathrm{d}t
\end{aligned}$$

For the first part, according to Lemma 3, since the error term is invariant with respect to translations on $\nabla \log p_t(\cdot)$ and the homogeneity of the ReLU neural network, we can omit $\nabla \log p_t(\mathbf{0}) = 0$ and rescale bound for the input data, for any $\varepsilon \in (0,1)$, there exist an approximation function $\overline{s}$ satisfying $\|\nabla \log p_t(\cdot) - \overline{s}(\cdot, t)\|_\infty \leq \varepsilon$, then we have:

$$\frac{1}{T - t_0} \int_{t_0}^{T} \|\nabla \log p_t(\cdot) - \overline{s}(\cdot, t)\|_{\ell^2(p_t)}^2 \, \mathrm{d}t \leq d\varepsilon^2 \,,$$

that satisfy the configuration of network architecture in Lemma 3.

For the second part:

$$\mathcal{L}(\overline{s}) - \frac{1}{T - t_0} \int_{t_0}^{T} \|\nabla \log p_t(\cdot) - \overline{s}(\cdot, t)\|_{\ell^2(p_t)}^2 \, \mathrm{d}t$$
$$= \frac{1}{T - t_0} \int_{t_0}^{T} \left( \mathbb{E}_{\boldsymbol{x}_0 \sim p_0} \mathbb{E}_{\boldsymbol{x}_t \sim p_{0t}(\boldsymbol{x}_t | \boldsymbol{x}_0)} \left[ \|\nabla_{\boldsymbol{x}_t} \log p_{0t}(\boldsymbol{x}_t | \boldsymbol{x}_0) - s(\boldsymbol{x}_t, t)\|_2^2 \right] - \|\nabla \log p_t(\cdot) - \overline{s}(\cdot, t)\|_{\ell^2(p_t)}^2 \right) \mathrm{d}t \,.$$

According to Vincent [2011], we have:

$$\mathbb{E}_{\boldsymbol{x}_0 \sim p_0} \mathbb{E}_{\boldsymbol{x}_t \sim p_{0t}(\boldsymbol{x}_t | \boldsymbol{x}_0 = \boldsymbol{x}_{(i)})} \left[ \left\| \nabla_{\boldsymbol{x}_t} \log p_{0t}(\boldsymbol{x}_t | \boldsymbol{x}_0 = \boldsymbol{x}_{(i)}) - s(\boldsymbol{x}_t, t) \right\|_2^2 \right] - \|\nabla \log p_t(\cdot) - \overline{s}(\cdot, t)\|_{\ell^2(p_t)}^2$$
$$= \mathbb{E}_{\boldsymbol{x}_0 \sim p_0} \mathbb{E}_{\boldsymbol{x}_t \sim p_{0t}(\boldsymbol{x}_t | \boldsymbol{x}_0)} \left[ \|\nabla_{\boldsymbol{x}_t} \log p_{0t}(\boldsymbol{x}_t | \boldsymbol{x}_0)\|_2^2 \right] - \|\nabla \log p_t(\cdot)\|_{\ell^2(p_t)}^2 \,,$$

which is an absolute value that does not depend on $s$. So we can define that:

$$E_2 := \mathbb{E}_{\boldsymbol{x}_0 \sim p_0} \mathbb{E}_{\boldsymbol{x}_t \sim p_{0t}(\boldsymbol{x}_t | \boldsymbol{x}_0)} \left[ \|\nabla_{\boldsymbol{x}_t} \log p_{0t}(\boldsymbol{x}_t | \boldsymbol{x}_0)\|_2^2 \right] - \|\nabla \log p_t(\cdot)\|_{\ell^2(p_t)}^2 \,.$$

So if we choose $\overline{s}$ is the approximation function that Lemma 3 provide, then we have:

$$\mathcal{L}(\overline{s}) \leq d\varepsilon^2 + E_2 \,.$$

**Putting things together**   Combine all three terms, we have:

$$\mathcal{L}(\hat{s}) \leq \left( \mathcal{L}(\hat{s}) - (1 + a)\hat{\mathcal{L}}(\hat{s}) \right) + (1 + a)\left( \hat{\mathcal{L}}(\overline{s}) - (1 + a)\mathcal{L}(\overline{s}) \right) + (1 + a)^2 \mathcal{L}(\overline{s})$$
$$\leq \left( \mathcal{L}(\hat{s}) - (1 + a)\hat{\mathcal{L}}(\hat{s}) \right) + (1 + a)\left( \hat{\mathcal{L}}(\overline{s}) - (1 + a)\mathcal{L}(\overline{s}) \right) + (1 + a)^2 (d\varepsilon^2 + E_2)$$
$$= \left( \mathcal{L}(\hat{s}) - (1 + a)\hat{\mathcal{L}}(\hat{s}) \right) + (1 + a)\left( \hat{\mathcal{L}}(\overline{s}) - (1 + a)\mathcal{L}(\overline{s}) \right) + (1 + a)^2 d\varepsilon^2 + (2a + a^2)E_2 + E_2$$

Then:

$$\frac{1}{T - t_0} \int_{t_0}^{T} \|\nabla \log p_t(\cdot) - \hat{s}(\cdot, t)\|_{\ell^2(p_t)}^2 \, \mathrm{d}t$$
$$= \mathcal{L}(\hat{s}) - E_2$$
$$= \left( \mathcal{L}(\hat{s}) - (1 + a)\hat{\mathcal{L}}(\hat{s}) \right) + (1 + a)\left( \hat{\mathcal{L}}(\overline{s}) - (1 + a)\mathcal{L}(\overline{s}) \right) + (1 + a)^2 d\varepsilon^2 + (2a + a^2)E_2$$
$$\lesssim \left( \frac{1 + 3/a}{n} \left( \frac{d(T - \log(t_0))}{T - t_0} + C_d^2 \right) \log \frac{\mathcal{N}_c(\tau, \mathcal{S})}{\delta} + (2 + a)\tau \right)$$
$$+ (1 + a)\left( \frac{1 + 3/a}{n} \left( \frac{d(T - \log(t_0))}{T - t_0} + C_d^2 \right) \log \frac{1}{\delta} + (2 + a)\tau \right)$$
$$+ (1 + a)^2 d\varepsilon^2 + (2a + a^2)E_2 \,,$$

with probability at least $1 - 2\delta - 2L\exp(-\Omega(m))$ over the randomness of initialization $\boldsymbol{W}$.

Let $a = \varepsilon^2$ and $\tau = \frac{1}{n}$, then we have:

$$\frac{1}{T - t_0} \int_{t_0}^{T} \|\nabla \log p_t(\cdot) - \hat{\boldsymbol{s}}(\cdot, t)\|_{\ell^2(p_t)}^2 \, \mathrm{d}t \lesssim \frac{1}{n\varepsilon^2} \left( \frac{d(T - \log(t_0))}{T - t_0} + C_d^2 \right) \log \frac{\mathcal{N}_c(\frac{1}{n}, \mathcal{S})}{\delta} + \frac{1}{n} + d\varepsilon^2 \,,$$

with probability at least $1 - 2\delta - 2L\exp(-\Omega(m))$ over the randomness of initialization $\boldsymbol{W}$. $\qquad\square$

## H  Discussion of Lipschitz property of score function

Here we provide an example to illustrate how the Lipschitz constant of the score function in a causal model is related to the model's nonlinear functions.

Here we give an example with $d = 3$, the causality is $x^{(1)} \Rightarrow x^{(2)} \Rightarrow x^{(3)}$.

According to Eq. (9), we have that:

$$s_1(\boldsymbol{x}) = -\frac{x^{(1)}}{\sigma_1^2} + \frac{\partial f_2(x^{(1)})}{\partial x^{(1)}} \frac{\epsilon_2}{\sigma_2^2}, \quad s_2(\boldsymbol{x}) = \frac{f_2(x^{(1)}) - x^{(2)}}{\sigma_2^2} + \frac{\partial f_3(x^{(2)})}{\partial x^{(2)}} \frac{\epsilon_3}{\sigma_3^2}, \quad s_3(\boldsymbol{x}) = \frac{f_3(x^{(2)}) - x^{(3)}}{\sigma_3^2} \,.$$

Then we can derive that:

$$\frac{\partial s_1(\boldsymbol{x})}{\partial x^{(2)}} = \frac{\partial s_1(\boldsymbol{x})}{\partial x^{(3)}} = \frac{\partial s_2(\boldsymbol{x})}{\partial x^{(3)}} = \frac{\partial s_3(\boldsymbol{x})}{\partial x^{(1)}} = \frac{\partial s_3(\boldsymbol{x})}{\partial x^{(2)}} = 0 \,,$$

$$\frac{\partial s_1(\boldsymbol{x})}{\partial x^{(1)}} = -\frac{1}{\sigma_1^2} + \frac{\partial^2 f_2(x^{(1)})}{\partial x^{(1)2}} \frac{\epsilon_2}{\sigma_2^2} \,,$$

$$\frac{\partial s_2(\boldsymbol{x})}{\partial x^{(2)}} = -\frac{1}{\sigma_2^2} + \frac{\partial^2 f_3(x^{(2)})}{\partial x^{(2)2}} \frac{\epsilon_3}{\sigma_3^2} \,,$$

$$\frac{\partial s_2(\boldsymbol{x})}{\partial x^{(1)}} = \frac{\partial^2 f_3(x^{(2)})}{\partial x^{(2)} \partial x^{(1)}} \frac{\epsilon_3}{\sigma_3^2} \,,$$

$$\frac{\partial s_3(\boldsymbol{x})}{\partial x^{(3)}} = -\frac{1}{\sigma_3^2} \,.$$

We denote $\boldsymbol{J}$ as the Jacobian of the score function. Then we can derive:

$$\|\boldsymbol{J}\|_{\ell_\infty} = \max \left( \left| -\frac{1}{\sigma_1^2} + \frac{\partial^2 f_2(x^{(1)})}{\partial x^{(1)2}} \frac{\epsilon_2}{\sigma_2^2} \right|, \left| -\frac{1}{\sigma_2^2} + \frac{\partial^2 f_3(x^{(2)})}{\partial x^{(2)2}} \frac{\epsilon_3}{\sigma_3^2} \right| + \left| \frac{\partial^2 f_3(x^{(2)})}{\partial x^{(2)} \partial x^{(1)}} \frac{\epsilon_3}{\sigma_3^2} \right|, \frac{1}{\sigma_3^2} \right)$$

$$\leq \max \left( \frac{1}{\sigma_1^2} + \left| \sup \frac{\partial^2 f_2(x^{(1)})}{\partial x^{(1)2}} \frac{\epsilon_2}{\sigma_2^2} \right|, \frac{1}{\sigma_2^2} + \left| \sup \frac{\partial^2 f_3(x^{(2)})}{\partial x^{(2)2}} \frac{\epsilon_3}{\sigma_3^2} \right| + \left| \sup \frac{\partial^2 f_3(x^{(2)})}{\partial x^{(2)} \partial x^{(1)}} \frac{\epsilon_3}{\sigma_3^2} \right|, \frac{1}{\sigma_3^2} \right) \,.$$

Then we have that for any $L$ satisfy:

$$L \geq \max \left( \frac{1}{\sigma_1^2} + \left| \sup \frac{\partial^2 f_2(x^{(1)})}{\partial x^{(1)2}} \frac{\epsilon_2}{\sigma_2^2} \right|, \frac{1}{\sigma_2^2} + \left| \sup \frac{\partial^2 f_3(x^{(2)})}{\partial x^{(2)2}} \frac{\epsilon_3}{\sigma_3^2} \right| + \left| \sup \frac{\partial^2 f_3(x^{(2)})}{\partial x^{(2)} \partial x^{(1)}} \frac{\epsilon_3}{\sigma_3^2} \right|, \frac{1}{\sigma_3^2} \right) \,,$$

then the $L$ is one of the Lipschitz constants of the score function.

According to the previous analysis, we can obtain the Lipschitz property of the score function by imposing some assumptions on the nonlinear function and noise of the model. For causal models with more complex DAG, the relationship between Lipshcitz and the model will be more complicated, but as long as the second derivatives of nonlinear functions are bounded and the variance of the noise is non-zero, the score function has Lipschitz property, and the value of the Lipschitz constant depends on the nonlinear function and noise of the model.

# I Broader Impacts

This is a theoretical work that provides theoretical analysis for causal inference based on score matching. As such, we do not expect our work to have negative societal bias, as we do not focus on obtaining state-of-the-art results in a particular task. On the contrary, our work can have various benefits for the community:

- Causal inference is crucial in fields such as medicine, social sciences, and economics for understanding the essence of phenomena and formulating effective intervention measures. The outcomes of this work not only provide researchers in these fields with more reliable and interpretable theoretical insights into causal inference, driving scientific advancements and societal development, but also the score matching-based causal inference methods can help uncover hidden causal effects and mechanisms, providing a scientific foundation for decision-making in areas such as social equity, educational policies, and medical interventions.

- The theoretical framework and methods developed in this work can inspire and inform other causal inference approaches, fostering interdisciplinary research and collaboration, and expanding the application scope of causal inference in different domains.