# OpenReview forum: "Sample Complexity Bounds for Score-Matching: Causal Discovery and Generative Modeling"
_NeurIPS.cc/2023/Conference — NeurIPS 2023 poster_

### Official Review · Reviewer_LFXr · 2023-06-28

**Soundness:** 3 good
**Presentation:** 2 fair
**Contribution:** 3 good
**Rating:** 4
**Confidence:** 3

**Summary:**

This work focus on theoretical analysis on two main score-based approaches. The first one is the score-based casual discovery, where the authors give the sample complexity error bounds for score matching using ReLU NNs as well as an upper bound of the error rate. The second one is the score-based generative modeling, where the authors give a sample complexity bound for the score estimation.

**Strengths:**

This work has strong theoretical results and has provided sufficient backgrounds for the readers to understand the theoretical problems and include a comprehensive discussion on related work.


**Weaknesses:**

- It is unclear why the analysis on score-based casual discovery and that on score-based generative modeling are put together in this work; it seems to me they are unrelated work. The authors might want to put some insights on the relationship between these two work, for example, some shared techniques for proving the sample complexity bounds.
- The definition of covering number should be included to make this work more self-contained.
- For Assumption 2, since this is an original assumption proposed by this work, the authors should give more justification to how much this assumption holds in practice. An specific example, even a simple one, showing when this assumption holds, would make this work more convincing.
- To provide some proof sketch might make this work more convincing and help reader better understand what are the technical contributions of this work. Now it is unclear to me why it provides better analysis than the previous work; specifically, why it does not rely on the assumption of low-dimensional data while the previous work Chen et al. [2023a] does.


**Questions:**

- It seems that the theoretical results of score-based casual discovery are highly specific to the algorithm in Rolland et al. [2022]. I wonder how these results generalize to other related approaches?
- Does Lemma 1 play a role in the later analysis? I wonder why we need Lemma 1.
- The assumption in Line 148-149 is not properly justified.
- Can the authors provide a specific case on when Assumption 2 holds and analysis on how practical this assumption?
- As mentioned above, why this work does not rely on the assumption of low-dimensional data while the previous work Chen et al. [2023a] does?

Minor issue:
- It seems that the case l = 1 in Equation (3) can be merged into the case 2<=l<=L-1.
- Line 107: donate -> denotes
- The property mentioned in Line 108 seems unrelated to the Definition 1.

**Limitations:**

Yes.

---

> ### Author Rebuttal · Authors · 2023-08-09
>
> We sincerely appreciate the feedback and suggestions provided by reviewer LFXr. Regarding the issues raised by the reviewer concerning the relationship between causal inference and the generative model aspects of the paper, as well as the convincing of assumption 2, we have addressed these matters in the [general response](https://openreview.net/forum?id=uNnPWR66b8&noteId=VWjUWDv2Kl). Furthermore, in the next version of the paper, we will present the main results in a more comprehensive and clearer way, emphasizing the key points of the paper and providing proof sketches to support the main results. Below, we will provide responses to additional questions raised by the reviewer. Additionally, we will address the details raised and incorporate the solutions in the next version of the paper.
>
> **Q1: The definition of covering number.**
>
>
> A1: The concept of the covering number plays a significant role in statistical learning theory. It is a widely used tool, often appearing in various learning theory textbooks. We appreciate the reviewer's suggestion. To enhance the self-contained nature of our paper, we plan to incorporate a new section in the appendix. This section will provide a comprehensive background definition of the covering number, ensuring that readers can grasp its essence without needing to refer to external sources.
>
> **Q2: Low-dimensional assumptions in previous work.**
>
>
> A2: The previous work [1] rested on the assumption of low-dimensional data structures, employing this to decompose the score function and engineer specialized network architectures for the derivation of the upper bound. Our work takes a distinct route. We harness the inherent traits and conventional techniques of standard deep ReLU networks to directly deduce the upper error bound. We will add such a discussion in the next version of the paper.
>
> **Q3: How do these results generalize to other related approaches?**
>
> A3: Firstly, our theoretical results in Theorems 1 and 3 are not influenced by the choice of algorithm used. Although Theorem 2 is established based on Algorithm 1, our technique provides a comprehensive processing strategy for those analogous algorithms which initially conduct a topological ordering based on the score function, followed by pruning to obtain the final DAG. So through suitable adaptations, our method can also be extended to similar algorithms [2][3].
>
> **Q4: The role of Lemma 1 in the paper.**
>
> A4: While Lemma 1 is not directly employed in the ensuing proofs presented in this paper, it serves as the cornerstone for Algorithm 1 in [2] in our analysis. Furthermore, it sheds light on vital properties of the non-linear additive Gaussian noise model. Its incorporation into the primary content might not be deemed essential, we plan to rectify this in the next version of the paper. We will introduce a dedicated section that delves into the discourse concerning Algorithm 1 and Lemma 1 in the appendix.
>
> **Q5: More discussion about Assumption 2.**
>
> A5: The left-hand side of Assumption 2 represents the expectation of the square of a random variable, this value is 0 when the function f is linear. Correspondingly, the value of $C_m$ is 0 too. For non-linear functions f, this value is greater than 0 (disregarding some extreme mathematical cases), and the corresponding $C_m$ is also greater than 0.
>
> Theorem 2 converges when $C_m$ is greater than 0, indicating that Algorithm 1 can converge for the causal discovery of any nonlinear causal relationships.
>
> Different non-linear functions f correspond to different values of $C_m$. The experimental results we provided in the general response demonstrate the varying outcomes of Algorithm 1 under causal relationships with different $C_m$ values.
>
> **Q6: The assumption in Line 148-149 is not properly justified.**
>
> A6: We disagree with the reviewer's view here. The content in lines 148-149 is not an assumption; it is based on the standard setting of theoretical analysis in Ornstein–Uhlenbeck process in score-based generative modeling, which has been widely employed in prior research [1, 4, 5]. However, we appreciate the reviewer's feedback and have added a remark at this point to clarify this setting.
>
> **References**
>
> [1] Score Approximation, Estimation and Distribution Recovery of Diffusion Models on Low-Dimensional Data. ICML 2023.
>
> [2] Scalable Causal Discovery with Score Matching. CLeaR 2023.
>
> [3] Causal Discovery with Score Matching on Additive Models with Arbitrary Noise. CLeaR 2023.
>
> [4] Sampling is as easy as learning the score: theory for diffusion models with minimal data assumptions. ICLR 2023.
>
> [5] Diffusion Schrödinger Bridge with Applications to Score-Based Generative Modeling. NeurIPS 2021.

---

> > ### Comment · Reviewer_LFXr · 2023-08-16
> >
> > I would like to thank the authors for the clarifications. Even though the relationship between the score-based casual discovery and score-based generative modeling is briefly explained in the general response, it would require non-trivial modification to the current presentation of this paper to address this concern and to include the new experimental results. Thus, I would keep my score.

---

> > > ### Author Response · Authors · 2023-08-17
> > >
> > > Dear reviewer LFXr,
> > >
> > > Thanks for the prompt response. We did indeed commit to making some revisions in the general response. But it's important to note that these revisions primarily focus on the organization of the paper and the explanation of the main results. These revisions do not affect the core theoretical results, major contributions, and novelty of the paper. Furthermore, the new experiment results serve to enhance the comprehensiveness of the paper and should not be regarded as a negative factor. Therefore, we hope the reviewer can reconsider their opinion.
> > >
> > > Best,
> > >
> > > Authors

---

> ### Author Response · Authors · 2023-08-15
> **Any remaining questions from reviewer LFXr?**
>
> Dear reviewer LFXr,
>
> As we're getting close to the end of the discussion period, we want to thank you once more for your time. We hope our comments so far have covered any past worries or questions, but if there's anything you'd like more explanation on, just tell us. If you've got any other questions, please don't hesitate to ask. We'll do our best to handle them before the deadline.

---

### Official Review · Reviewer_FzFv · 2023-07-03

**Soundness:** 4 excellent
**Presentation:** 4 excellent
**Contribution:** 4 excellent
**Rating:** 7
**Confidence:** 4

**Summary:**

The paper provides sample complexity bounds on score function estimation when (1) the score function is estimated by using SGD to minimize a denoising score matching objective, (2) the probability distribution is induced by a structural causal model (SCM) with additive Gaussian noise, and (3) the score function is Lipschitz, and (4) for each mechanism $f_i$ in the SCM, the expected value of the second derivative of $f_i$ with respect to each parent of $X_i$ is lower-bounded by $C_m$ times $\sigma_i$, the variance of the exogenous noise of $X_i$.

They show that, under these conditions, the estimated score function converges at a parametric rate (Theorem 1). Using this bound, they give a sample complexity bound for the event that the SCORE algorithm for causal order search returns a correct topological order (Theorem 2). Finally, they also use the bound from Theorem 1 to provide a sample complexity result for the score-matching objective used in those works.

**Strengths:**

### Significance
The paper appears to be a significant theoretical contribution addressing the sample complexity of score function estimation, which as an important topic for both causal structure learning and score-based generative modeling.

### Clarity
The paper is written in a fairly clear style. Relevant notations are appropriately defined and summarized in the Appendix. For each theorem, there are accompanying remarks which clearly discuss the implications of the theorem. The motivation and the context for the work are easy to understand.

**Weaknesses:**

### Experimental Results
This is clearly a theory paper, and contributes enough theory such that there is no need for an extended section on experiments. In fact, given the space constraints, extensive experiments would most likely hurt the paper. However, a small set of experiments - e.g. corroborating Theorem 1 by plotting the loss function versus the number of samples - would take the paper from a good contribution to a great one (8 instead of 7). In my experience, experimental validation of theoretical results on sample complexity can be fairly difficult to obtain, and thus papers like this one would be a lot more useful in practice if they ran some carefully-chosen experiments.

**Questions:**

### Details
1. The theorems (eg. Theorem 1 and Theorem 3) refer to a DNN trained by SGD. However, this is missing a condition on the learning rate. What is the condition on the learning rate?
2. Presumably the usage of "SGD" in the theorems here implies that batch size = 1. Do the results extend to larger batch sizes? For clarity, I think it would be helpful to encapsulate the assumptions on the architecture and training into an Assumption environment.
3. What is $\tau$, the first argument to the covering number in line 203? Where is it defined?

### Suggestions
1. Since this paper sits at the intersection of score-based generative modeling (SGM) and causal structure learning (CSL), it is important to keep the background of both audiences in mind. In particular, coming from causal structure learning / statistics, I am surprised by Theorem 1, which seems to require that SGD finds a solution close to the global optimum. It looks like Theorem 1 uses some relatively recent results / techniques (e.g. Nguyen et al., 2021), which might be well-known in the SGM community but are not well-known in the CSL community. This paper provides a good opportunity to bring the advances of SGM to the CSL audience, and it would be valuable to provide more intuition about the techniques used here.
2. Please reference all appendices in the paper. Appendices B, C, F, and G are not referenced in the paper.
3. I think Lemma 2 would be more clear if you left out the node $i$ altogether. Since the associated term is zero anyways, it just gets in the way and serves no purpose in the left-hand side of the implication statement. It feels like the point that the variance term associated to $i$ is zero shouldn't be part of this theorem, but a separate observation.

**Limitations:**

The authors have adequately addressed the limitations of their work.

---

> ### Author Rebuttal · Authors · 2023-08-09
>
> We deeply appreciate the feedback and suggestions from reviewer FzFv. Regarding the issues raised by the reviewer regarding the relationship between causal inference and the generative model aspects of the paper, as well as the lack of experimental validation, we have provided responses in the [general response](https://openreview.net/forum?id=uNnPWR66b8&noteId=VWjUWDv2Kl). We will also enhance the comprehensiveness and clarity of the main results in the next version of the paper, emphasizing the paper's focal points. We will answer the other questions from the reviewer below. Furthermore, for the detailed problems, we will resolve them and incorporate the solutions in the next version of the paper.
>
> **Q1: What is the condition on the learning rate and can we extend the result to a larger batch GD.**
>
>
> A1: The use of SGD training is referenced in our three theorems. However, it's important to note that Theorem 1 and Theorem 3 are rooted in the generalization by sampling complexity bound. This makes them independent of the specific algorithm used, and consequently, we did not delve into learning rates here. The results are broadly applicable and can be seamlessly extended to encompass larger batch GD.
>
> Regarding Theorem 2, its foundation lies in the proof of SGD/GD convergence within deep networks, as elucidated in [1]. This proof necessitates certain conditions related to large network width ($m\geq\text{poly}(n,L)$) and a sufficiently small learning rate ($\mathcal{O}(\frac{1}{\text{poly}(n,L)m \log^2 m})$). These convergence outcomes also apply to BatchGD, as demonstrated in similar convergence results [2]. Hence, Theorem 2 can naturally be expanded to incorporate Batch GD as well.
>
> **Q2: The definition of $\tau$ in the covering number.**
>
> A2: Sorry for the typo here. Instead of $\tau$, it should be $\frac{1}{n}$. We have revised this issue.
>
> **Q3: Modifications for Lemma 2.**
>
> A3: We presented Lemma 2 in this form to facilitate its utilization within the proof. But we concur with the reviewer's perspective that leaving out the reference to node $i$ in this lemma could enhance its clarity. As a result, we have restructured the phrasing of this lemma and subsequently refined the associated proof.
>
> **References**
>
> [1] A Convergence Theory for Deep Learning via Over-Parameterization. ICML 2019.
>
> [2] Convergence rates for gradient descent in the training of overparameterized artificial neural networks with biases.

---

> > ### Comment · Reviewer_FzFv · 2023-08-15
> >
> > I thank the others for their thoughtful response. Their responses to **Q2** and **Q3** have satisfied my concerns on those points.
> >
> > With regard to **Q1**: could your please describe how your response will be incorporated into the paper? e.g. it is my understanding that the mention of SGD in Theorem 1 and Theorem 3 can be removed, while for Theorem 2, the condition needs to be added? In particular, the condition could be formally defined in the appendix and the extension to batch GD could be added as a remark.

---

> > > ### Author Response · Authors · 2023-08-17
> > >
> > > Dear reviewer FzFv,
> > >
> > > We thank the reviewer for the positive reply. As the reviewer said the SGD mentioned in Theorem 1 and Theorem 3 can indeed be deleted, but in order to maintain the unity of the paper, we will keep it and discuss this point in the Remark, and we will discuss Theorem 2 according to your suggestion to add discussion in the appendix.
> > >
> > > We are happy that our clarifications addressed most of your concerns and please feel free to let us know if you have any remaining concerns or further questions.
> > >
> > > Best,
> > >
> > > Authors

---

### Official Review · Reviewer_RLDi · 2023-07-05

**Soundness:** 3 good
**Presentation:** 2 fair
**Contribution:** 2 fair
**Rating:** 6
**Confidence:** 2

**Summary:**

This study aims to investigate the sample complexity associated with score-matching and its applications in the field of causal discovery, providing valuable theoretical bounds. The authors present theoretical evidence that training a conventional deep ReLU neural network using stochastic gradient descent enables the accurate estimation of the score function. Additionally, they establish rigorous bounds on the error rate pertaining to the recovery of causal relationships, employing the score-matching-based methodology introduced by Rolland et al. [2022], under the assumption of an adequately precise estimation of the score function. Furthermore, an analysis is performed to determine the upper bound of score-matching estimation within the framework of score-based generative modeling, which not only bears relevance to causal discovery but also demonstrates independent significance within the wider domain of generative models.

**Strengths:**

There are several main strengths in this paper. First, the authors extensively explored the relevant recent theories, including background research. Second, the error bounds are rigorously derived from well-known assumptions (1 and 3) and novel assumption (2) for causal discovery. Third, the use of this theory seems to have potential in multiple applications relating to causal discovery and SGM given that the architecture is general enough.

**Weaknesses:**

There are some weaknesses in this paper. First, the background section is a bit too long compared to the main body (almost 5 pages with added related work in Section 5) resulting in only 2.5 pages new materials (excluding conclusion). Although it was advantageous to extensively explain the background (where there are multiple topics in this paper), it became a weakness due to its excessive length.

The second weakness is about deep ReLU neural networks, but there is no empirical experiments to corroborate theory. For example, one might show causal discovery errors and how the bounds in Theorem 2 are ‘good’ enough. If the bounds are too loose, the usefulness of the bounds will become low. (or how robust the result would be under the violation of the some of assumptions)

Given that there are some space that can be saved by removing (or moving) some of related work/preliminaries from main text, I recommend the authors to have some proof-of-concept experiments demonstrating the effectiveness of such theoretical bounds.


**Questions:**

Not a question but there are issues with \begin{align} or other tex related comments. If you have an empty line in between text and mathematical formula, there will be a gap. You can see Eq 1 (and the one before and one next), 4, 5, 6, 7, (one next to 7), 8, next to 8, in Assumption 2, in Theorem 1, in Thm 2, in Thm 3. You can effectively save lots of space.

There are many remarks. But I guess they can be incorporated more smoothly just as a text with proper rephrasing (if needed).

98th line contains a typo with missing parentheses in the equation.

Theorem 3, Remark 1. Can you elaborate in the main text what enable your theorem applicable to high-dimensional data versus. Chen 2023a's results on low dimensional data?

**Limitations:**

Assumptions can be restrictive but the authors are well-understanding such restrictions and others also made similar assumptions. For the assumption 2, it is unclear how one can assume C_m properly.

---

> ### Author Rebuttal · Authors · 2023-08-09
>
> We greatly appreciate the comments and suggestions provided by reviewer RLDi. Regarding the convincing of Assumption 2 and the lack of validation experiments in the paper, we have addressed these concerns in the [general response](https://openreview.net/forum?id=uNnPWR66b8&noteId=VWjUWDv2Kl). In the next version of the paper, we will condense the background part, emphasize the key points of the paper, and provide proof sketches for the main results to present them in a more concise and clearer way. We will now provide responses to the other questions raised by the reviewer. Additionally, for the detailed issues, we will address them and incorporate the changes into the next version of the paper.
>
> **Q1: Low-dimensional assumptions in previous work.**
>
> A1: The previous work [1] rested on the assumption of low-dimensional data structures, employing this to decompose the score function and engineer specialized network architectures for the derivation of the upper bound. Our work takes a distinct route. We harness the inherent traits and conventional techniques of standard deep ReLU networks to directly deduce the upper error bound. We will add such a discussion in the next version of the paper.
>
> **References**
>
> [1] Score Approximation, Estimation and Distribution Recovery of Diffusion Models on Low-Dimensional Data. ICML 2023.

---

> > ### Comment · Reviewer_RLDi · 2023-08-12
> >
> > Thank you for your response. The authors mentioned revising the paper based on reviewers comments (not just mine). Hence, I am positive about what the paper will look like. Yet my score will remain unchanged (weak accept) given my limited confidence level.

---

> > > ### Author Response · Authors · 2023-08-17
> > >
> > > Dear reviewer RLDi,
> > >
> > > Thanks for your insightful comments. Please let us know if you have any further questions.
> > >
> > > Best,
> > >
> > > Authors

---

### Official Review · Reviewer_Ndh6 · 2023-07-06

**Soundness:** 3 good
**Presentation:** 2 fair
**Contribution:** 3 good
**Rating:** 4
**Confidence:** 2

**Summary:**

This paper presents error bounds (sample complexity and convergence rates) for the problems of score based generative modelling and score-order based causal discovery. It is primarily a theory paper, and one of the first few works to present error bounds for a causal discovery algorithm that is not conditional independence based.

**Strengths:**

(+) In score-function based causal discovery (which does not use conditional independence testing), there are no notable error bounds. So this work addresses the gap. Although it is tied to the order based method of Rolland et al, it is interesting as this might lead to more such work for other methods in causal discovery. Theoretical guarantees are important for causal discovery as real world validation is harder due to lack of ground truth.
(+) The assumptions and technicalities are fairly well laid out. Although the score requires Lipschitz assumptions, thereby limiting the class of mechanisms, it is still relevant as it still might cover fairly large class of real world settings.
(+) The theoretical analysis for score based generative modelling seems similar to what has been done for causal discovery, and is potentially interesting. However, I am unsure about its relevance wrt that field as I am not totally familiar with it.

**Weaknesses:**

(-) My main concern with this work is that it might be a bit too broad in scope. It is definitely okay and also welcome to have a broader scope (causal discovery plus score based generative modelling), but I don't think the current writing/presentation justifies it appropriately. I think a better way to present the results would have been to present the main result a bit more generally and clearly, and explain that result in the context of both causal discovery and score based generative modelling. I would have been happy to see just the causal discovery part alone as it merits quite a bit of discussion (since it is the first work doing so).
(-) There are no experiments. Although it is primarily a theory paper, some simple empirical analysis of RELU deep networks for score function in causal discovery would have been very illustrative and relevant. It would have been good to see how the bounds when the assumptions are satisfied and how do they get worse if the assumptions are not satisfied.

**Questions:**

My main question is : Is there a common theoretical part (proof techniques and statements) that can be generalised to both settings? If so, I think it might be good to highlight it. If not, I feel the paper might benefit from studying these two problems separately.

**Limitations:**

The assumptions are fairly well laid out. A major limitation now is that there is no empirical analysis.

---

> ### Author Rebuttal · Authors · 2023-08-09
>
> We sincerely appreciate the input and suggestions provided by reviewer Ndh6. Regarding the issues raised by the reviewer concerning the relationship between causal discovery and the generative model of the paper, as well as the lack of experimental validation, we have addressed these concerns in the [general response](https://openreview.net/forum?id=uNnPWR66b8&noteId=VWjUWDv2Kl). We will also present the main results in a more comprehensive and clearer way in the next version of the paper, highlighting the main points of the paper.

---

> ### Author Response · Authors · 2023-08-15
> **Any remaining questions from reviewer Ndh6?**
>
> Dear reviewer Ndh6,
>
> As we're getting close to the end of the discussion period, we want to thank you once more for your time. We hope our comments so far have covered any past worries or questions, but if there's anything you'd like more explanation on, just tell us. If you've got any other questions, please don't hesitate to ask. We'll do our best to handle them before the deadline.

---

> > ### Comment · Reviewer_Ndh6 · 2023-08-18
> >
> > Thanks a lot for the clarifications and I acknowledge your efforts to get experimental results. I am unsure if similar well tailored experiments in SGM setting would also be insightful as I am more coming from causal structure learning background. I still think the current theory and causal structure experiments is a positive contribution already.
> >
> > The major problem I see is that the proposed reorganization and presentation, which includes the current set of experiments, would involve significant and/or non-trivial changes to the overall paper (as the other reviewer pointed out too). It is also a bit less concrete at a lower level on how the paper would look like after all the changes take place. As a result, I will keep my current score under the conclusion that the paper might benefit from another round of review with all the changes appropriately incorporated.

---

### Official Review · Reviewer_wqRq · 2023-07-19

**Soundness:** 3 good
**Presentation:** 3 good
**Contribution:** 2 fair
**Rating:** 6
**Confidence:** 3

**Summary:**

This paper provides detailed theoretical results for causal discovery and score-based generative modeling. For causal discovery, it first provides a sample complexity analysis for the estimation of the score function for nonlinear additive Gaussian noise models, then it proves that the error rate of a score-matching-based method, SCORE, converges linearly w.r.t the number of training data. For score-based generative modeling, it presents sample complexity bounds for the estimation of the score function in the ScoreSDE.

**Strengths:**

1. This paper provides sound and detailed theoretical results for score matching in the context of both causal discovery and score-based generative modeling. In both cases, the theoretical analyses hold with mild assumptions compared to previous work.

2. This paper well organized overall. It provides a detailed discussion about preliminaries in Section 2, provides theoretical results for causal discovery and score-based generative modeling in Section 3 and Section 4 respectively. Although there are many symbols, I think this paper is relatively easy to follow.

3. This paper points out limitations of the current theoretical results clearly in Section 6.

**Weaknesses:**

1. As demonstrated in Section 6 of this paper, the theoretical results are still not general enough due to some relatively strong assumptions.

2. I suggest the authors change the title "Error Bounds for Score Matching Causal Discovery" because almost half of this paper is about score matching in score-based generative modeling, which has little to do with causal discovery.

3. Although I have no doubt about the soundness of the provided theoretical results, the contribution of this paper is somewhat limited. This paper does not design more advanced algorithm, it only provides sample complexity bounds for existing algorithms under PAC framework. I'm not sure whether it should be regarded as a supplement for previous work or an independent research. However, considering that ScoreSDE and is widely used in respective field, I still recommond "weak accept" for this paper.


**Questions:**

N/A

**Limitations:**

See weakness.

---

> ### Author Rebuttal · Authors · 2023-08-09
>
> We greatly appreciate the comments and suggestions from reviewer wqRq. Regarding the questions raised by the reviewer about the relationship between causal discovery and the generative model of the paper, as well as concerns about strong assumptions, we have provided responses in the [general response](https://openreview.net/forum?id=uNnPWR66b8&noteId=VWjUWDv2Kl). Below is the response addressing your additional questions:
>
> **Q1:  This paper does not design a more advanced algorithm.**
>
> A1: In this paper, we introduce the inaugural sample complexity bound for causal discovery in non-linear Gaussian models (to the best of our knowledge). This is an example of theory following practice, with the SCORE algorithm only having identifiability guarantees in the infinite data regime, but being backed up by empirical results with finite data sets. While we anticipate that this paper will establish a path toward statistically efficient causal discovery algorithms, we maintain that formulating a new algorithm falls beyond the scope of our present endeavor.

---

> > ### Comment · Reviewer_wqRq · 2023-08-12
> > **Response to rebuttal**
> >
> > Thank the authors for their work, I think the title "Sample Complexity Bounds for Score-Matching: Causal Discovery and Generative Modeling" is better than the original one. Besides, although I appreciate the theoretical contributions of this work, IMHO, the overall contributions may be somewhat insufficient, because it only derives sample complexity under the widely-used PAC framework. After rebuttal, I decide to maintain my score.

---

> > > ### Author Response · Authors · 2023-08-17
> > >
> > > Dear reviewer wqRq,
> > >
> > >
> > > Thanks for your insightful comments. Please let us know if you have any further questions.
> > >
> > >
> > > Best,
> > >
> > > Authors

---

### Author Rebuttal · Authors · 2023-08-09

# General response:

We extend our gratitude to the reviewers for their valuable feedback. In response to recurring issues highlighted by multiple reviewers, we offer a consolidated response as follows:

**Q1:  Relationship between two parts.**

A1: The theoretical foundation of this paper spans two distinct yet interconnected domains: causal discovery and score-based generative modeling. These two domains share intriguing interrelations while also exhibiting notable disparities.

These two domains are connected by a common theoretical foundation centered on the upper bound of score matching. Specifically, Theorems 1 and 3 study similar problems in different domains and share the same techniques drawn from statistical learning theory and deep learning theory.

However, they also harbor autonomous elements, with the findings in the causal discovery domain having a relatively broader and more consequential scope. The foundation of Theorem 2 rests upon Theorem 1, but its result pertains to entirely separate problems. It can be seen as an embodiment of applying the upper bound of score matching for causal discovery.

Consequently, we simplified the background part of the paper, relocating certain components to supplementary materials. Additionally, we enhance the explanations and provide succinct proof sketches for the theorem in the causal discovery part. This endeavor aims to present the main results in a manner universally comprehensible and illuminating.

Furthermore, if the reviewers think the title "Sample Complexity Bounds for Score-Matching: Causal Discovery and Generative Modeling" be better aligned with the paper's essence, we are willing to modify the title.

**Q2:  Lack of experiments.**

Following the reviewers' recommendations, we conducted a series of experiments to validate the theoretical findings presented in the paper. We took inspiration from the code provided in [1] and employed the structural Hamming distance (SHD) between the generated output and the actual causal graph to assess the outcomes. The ensuing experimental outcomes for SHD, vary across causal model sizes (d), sample sizes (n), and $C_m$. The specific results are shown in the pdf.

Analyzing the experimental outcomes, we find a notable pattern: higher values of $C_m$, augmented sample sizes $n$, and reduced model size $d$ all contribute to the algorithm's performance. This alignment with the insights from Theorem 2 in our paper is clear.

In the subsequent paper revision, we will incorporate these experimental findings to further enrich the paper's comprehensiveness.

**Q3:  Discussion of the assumptions.**

A3: As the reviewers point out, our assumptions are reasonable, with 1 and 3 being standard in the deep learning theory. Regarding assumption 2, we remark that this assumption is strongly linked with identifiability and that the algorithm does not make use of $C_m$.

This second assumption is particularly interesting because it relates the sample complexity bound with classical identifiability theory: if the mechanisms are linear, even with infinite data the graph is not identifiable. In this sense, this assumption cannot be violated.

An interesting avenue for future work would be the effect of local assumption violations i.e., what if only one mechanism is linear but others are not? We can add a discussion on this for future work, however, this is far beyond the scope of the present work. In fact, most identifiability results in the infinite data regime still require causal assumptions to hold globally.

**References**

[1] Score Matching Enables Causal Discovery of Nonlinear Additive Noise Models. ICML 2022.

---

### Decision · Program_Chairs · 2023-09-21

**Decision:**

Accept (poster)

**Comment:**

This paper offers comprehensive theoretical findings concerning both causal discovery and score-based generative modeling. In the realm of causal discovery, it initiates with an in-depth examination of sample complexity, specifically focusing on the estimation of the score function for nonlinear additive Gaussian noise models. Additionally, the paper establishes a proof demonstrating that the error rate of the score-matching-based method, known as SCORE, exhibits linear convergence with respect to the quantity of training data. In the context of score-based generative modeling, the paper also presents sample complexity bounds for the estimation of the score function within the ScoreSDE framework.

The paper addresses a timely analysis of a recent score matching based causal discveory method. It offers a rigorous theoretical analysis of the properties of score-matching-based causal discovery. All the reviewers acknowledge the theoretical contribution of this work. However, the major issues of the paper is that the current version needs a significant revision to account for the comments in the review comments and rebuttal. I recommend recommend acceptance of this paper due to its solid contribution. The authors should carefully revise the paper by taking into the review comments.